# Bioorganic Chemistry, Toxinology, and Pharmaceutical Uses of *Datura* Metabolites and Derivatives

**DOI:** 10.3390/toxins17090469

**Published:** 2025-09-18

**Authors:** Amin Mahmood Thawabteh, Saleh Sulaiman, Ilaf Omar Alabed, Laura Scrano, Donia Karaman, Rafik Karaman, Sabino A. Bufo

**Affiliations:** 1Department of Chemistry, Birzeit University, West Bank, Ramallah 00972, Palestine; athawabtah@birzeit.edu (A.M.T.); ssuliaman@birzeit.edu (S.S.); ilafomar6@gmail.com (I.O.A.); 2Department for Humanistic, Scientific and Social Innovation, University of Basilicata, 85100 Potenza, Italy; laura.scrano@unibas.it; 3CNR-IRSA Istituto di Ricerca Sulle Acque, 74121 Taranto, Italy; 4Pharmaceutical Sciences Department, Faculty of Pharmacy, Al-Quds University, Jerusalem 20002, Palestine; kdonia65@yahoo.com (D.K.); dr_karaman@yahoo.com (R.K.); 5Department of Basic and Applied Sciences, University of Basilicata, 85100 Potenza, Italy; 6Department of Geography, Environmental Management and Energy Studies, University of Johannesburg, Auckland Park Kingsway Campus, Johannesburg 2092, South Africa

**Keywords:** *Datura* alkaloids, muscarinic receptor antagonists, neuropharmacology, hyoscyamine, scopolamine, datumetine, atropine, tropine

## Abstract

*Datura* species have been recognized for their potent pharmacological properties, producing a diverse array of tropane and non-tropane alkaloids with significant clinical and toxicological relevance. This review synthesizes current knowledge on the biosynthesis, pharmacology, and therapeutic applications of 43 compounds isolated from *Datura*, with emphasis on both major constituents—such as atropine, hyoscyamine, and scopolamine—and minor alkaloids, including anisodamine, apoatropine, and datumetine. These alkaloids were classified into four significant categories, drawing on recent advances in plant biochemistry and analytical chemistry. The analysis is based on 204 peer-reviewed scientific publications from the past decade (2015–2025), highlighting both traditional ethnobotanical knowledge and recent pharmacological advances. The review details their enzymatic pathways, mechanisms of action at muscarinic and other receptor systems, pharmacokinetics, and dose-dependent toxicological profiles. Particular attention is given to lesser-studied derivatives and metabolites with emerging therapeutic potential, as well as their role in metabolic engineering, drug discovery, and forensic analysis. Notably, datum tine is highlighted for its unique NMDA receptor modulatory effects and neurotoxic potential, while tropine and hygrine serve as critical biosynthetic intermediates and analytical markers. By integrating biochemical, pharmacological, and toxicological insights, this work provides a comprehensive framework for future exploration of *Datura* alkaloids as both therapeutic agents and research tools.

## 1. Introduction

Members of the genus *Datura* (family Solanaceae) have fascinated humanity for millennia, owing to their potent pharmacological and toxicological properties. Across diverse cultures, *Datura* extracts were employed in traditional medicine and ritualistic practices, and as natural insecticides [1,2]. Beyond these historical uses, recent research has revealed a broader spectrum of biological activities for *Datura* species. Over the last decade, extracts and purified metabolites have demonstrated antioxidant, anti-inflammatory, antimicrobial (antibacterial, antifungal, antiviral), cytotoxic and anticancer, antidiabetic, analgesic, wound-healing, and nematocidal effects. Particular species also exhibit neuroprotective or neurotoxic actions, depending on dose and specific alkaloid composition. Multiple studies have reported larvicidal and insecticidal properties against agricultural pests and disease vectors, supporting the potential integration of this compound into pest management strategies. Notably, *D. metel* has been shown to possess bioherbicidal activity: controlled experiments demonstrated that this plant stimulates *Capsicum annuum* (agronomic crop) while it suppresses its associated weed, *Solanum elaeagnifolium*, likely through allelopathic mechanisms, suggesting a role in sustainable weed management [3]. More recently, Masum et al. (2023) reported that *D. metel* extract significantly reduces root and shoot lengths of *Parthenium hysterophorus*, underscoring its practical allelopathic potential in agricultural settings [4].

The diverse bioactivities of *Datura* are primarily attributable to its rich secondary metabolite profile. Tropane alkaloids—including atropine, hyoscyamine, and scopolamine—underpin anticholinergic, neurological, and bronchorelaxant effects, whereas withanolides, steroidal lactones, phenolics, and amide metabolites have been implicated in antioxidant, anti-inflammatory, cytotoxic, and antimicrobial actions. A growing number of metabolomic surveys are reporting non-tropane ingredients, such as β-carbolines and minor alkaloids, which may also contribute to the observed activities. Activity profiles, however, differ significantly depending on the species, plant organ, stage of development, extraction technique, and place of origin. Major tropane alkaloids have strong anticholinergic effects; thus, their use in medicine or agriculture necessitates careful dose optimization and safety assessment [3,4]. Modern phytochemical investigations have revealed that the characteristic bioactivity of these plants derives chiefly from their repertoire of tropane alkaloids—secondary metabolites that originally evolved as chemical defenses against herbivory and microbial attack [1,2]. Among these, atropine, hyoscyamine, and scopolamine have achieved enduring clinical importance, serving as life-saving antidotes in organophosphate poisoning, critical tools in anesthetic practice, and mainstays in ophthalmology and respiratory therapy [1,2].

Over the past two decades, advances in analytical chemistry—particularly ultra-high-performance liquid chromatography coupled with high-resolution mass spectrometry and two-dimensional NMR spectroscopy—have uncovered a wealth of both major and minor alkaloids in *Datura* tissues—these range from ubiquitous tropane esters to more elusive pyrrolidine and β-carboline derivatives [5,6]. Concurrently, recombinant expression of plant cytochrome P450 monooxygenases and acyltransferases has begun to elucidate the enzymatic cascades that convert simple amino acid precursors (e.g., ornithine and phenylalanine) into complex bicyclic and tricyclic frameworks [5,6]. Such pathway discoveries not only deepen our understanding of plant specialized metabolism but also open avenues for the metabolic engineering of high-value compounds and the production of novel semi-synthetic analogs [7,8].

Clinically, the utility of *Datura*-derived alkaloids extends beyond their historical roles. Primary tropane alkaloids underpin emergency protocols for severe cholinergic crises, while semi-synthetic quaternary ammonium derivatives (e.g., ipratropium, tiotropium) represent optimized bronchodilators with minimal central side effects [8]. Handy functions are provided by minor alkaloids like datumetine (an NMDA modulator with preclinical neurotoxicity), littorine (a direct metabolic precursor with a 35% conversion to hyoscyamine), and cuscohygrine (a forensic biomarker in the differentiation of cocaine from coca leaf). Furthermore, next-generation treatments that precisely target particular muscarinic receptor subtypes are anticipated when structure–activity relationship (SAR) investigations are integrated with emerging drug-delivery systems [9,10,11].

Despite these strides, significant gaps persist in our understanding of *Datura* metabolism, toxicological thresholds, and the structure–activity relationships that govern both therapeutic efficacy and adverse effects. In particular, the minor and less common alkaloids warrant systematic evaluation for potential applications in infectious disease, neuropharmacology, and beyond [10,11,12].

This review addresses these challenges by categorizing *Datura* metabolites into four primary categories and surveying cutting-edge methodologies for elucidating pathways and optimizing compounds. It concludes by presenting a roadmap: (i) engineering of Datura P450s in Nicotiana for alkaloid biosynthesis, (ii) leveraging receptor–ligand docking to refine subtype-selective muscarinic antagonists, and (iii) implementing human-relevant toxicology studies to set precise therapeutic windows for both established agents (atropine, scopolamine) and underexplored leads (anisodine, datumetine).

## 2. Classification of Alkaloids and Derivatives from *Datura* Species

To capture the full breadth of chemical architectures and bioactivities found in *Datura* species, a four-tiered classification framework was adopted that explicitly separates natural metabolites (core tropane alkaloids, minor alkaloids, and other natural alkaloids) from synthetic and semi-synthetic derivatives. First, the primary tropane alkaloids comprise the most abundant and intensively studied compounds—atropine, hyoscyamine, scopolamine, anisodamine, cuscohygrine, littorine, and datumetine—each arising directly from the core tropane skeleton [13,14,15]. Second, the minor and less common alkaloids (e.g., hygrine, apoatropine, anisodine, tropine) occur at trace levels yet offer intriguing variations on the tropane scaffold that may harbor novel bioactivities [16,17], as shown in Figure 1.

The third category encompasses synthetic and semi-synthetic derivatives, in which medicinal chemistry has leveraged the tropane core to produce quaternary ammonium and other analogs, such as ipratropium, tiotropium, methscopolamine, trospium, homatropine, tropicamide, cyclopentolate, oxybutynin, tolterodine, and trihexyphenidyl, with optimized pharmacokinetics and receptor selectivity. Finally, other alkaloid classes (notably β-carbolines such as harmane and norharmane, along with cuscohygrine variants) extend beyond the classical tropane series, underscoring the chemical complexity of *Datura* secondary metabolism [18,19]. In the sections that follow, each class is examined in turn, with attention to structural features, biosynthetic origin, natural abundance, and known pharmacological profiles.

### 2.1. Core Tropane Alkaloids in Datura

#### 2.1.1. Atropine

Atropine (compound **1** in Figure 2) is a tropane alkaloid predominantly isolated from species of the genus *Datura*, especially *Datura stramonium* L. and *Datura metel* L. It exists naturally as the racemate of (–)-hyoscyamine and (+)-hyoscyamine (d-hyoscyamine), the levorotatory enantiomer being the biologically active form [20,21]. In plants, atropine and its precursors serve as defensive secondary metabolites that deter herbivory due to their potent antimuscarinic activity [21,22]. In humans, atropine’s ability to competitively antagonize muscarinic acetylcholine receptors (mAChRs) has been harnessed for diverse therapeutic applications, ranging from the emergency management of organophosphate poisoning to preanesthetic preparation and as a long-acting ophthalmic cycloplegic, while its toxicity profile demands careful dosing and monitoring [22].

Chemical Structure and Stereochemistry

Structurally, it comprises a bicyclic tropane nucleus —7-azabicyclo[3.2.1]octane— with a tertiary amine at the bridgehead (N-8) and an ester linkage at C-3 to tropic acid (3-hydroxy-2-phenylpropanoic acid). Its full IUPAC name is (±)-(8-methyl-8-azabicyclo[3.2.1]octan-3-yl) 3-hydroxy-2-phenylpropanoate [23]. Three stereogenic centers exist at C-3, C-5, and C-6 of the tropane core. In pure (–)-hyoscyamine (the active enantiomer), these centers adopt the configuration (–)-(3S,5R,6S). For atropine (the racemic mixture), the C-3 center is present as a 1:1 mixture of (R)- and (S)-configurations, each paired with correspondingly inverted orientations at C-5 and C-6. Racemization from (–)-hyoscyamine to atropine can occur spontaneously under certain storage conditions, such as extreme pH levels, elevated temperatures, or during extraction processes [23,24].

The tropane alkaloid biosynthesis pathway in *Datura* spp. Initiates from L-ornithine (or arginine), which undergoes decarboxylation to putrescine via ornithine decarboxylase (ODC; EC 4.1.1.17). Putrescine is N-methylated by putrescine N-methyltransferase (PMT; EC 2.1.1.53) to N-methylputrescine. N-methylputrescine is oxidatively deaminated by N-methylputrescine oxidase (MPO; EC 1.4.3.10) to 4-methylaminobutanal, which cyclizes nonenzymatically to the N-methyl-Δ^1^-pyrrolinium cation—two successive Mannich-like condensations with acetate lead to hygrine, which then rearranges to tropinone. Tropinone is stereospecifically reduced by tropinone reductase I (TRI; EC 1.1.1.206) to tropine (3α-hydroxytropane). Tropine is esterified with tropic acid (derived from phenylalanine via phenylpyruvate and phenyllactic acid) by littorine synthase (a tropine:phenyllactate acyltransferase) to form littorine. Hyoscyamine 6β-hydroxylase (H6H; CYP80F1) then catalyzes a two-step oxidation of littorine: first generating hyoscyamine aldehyde, then reducing to (–)-hyoscyamine. Under physiological or extraction conditions, partial epimerization at C-3 produces (±)-atropine [25,26].

Mechanism of Action at the Molecular Level

The G protein–coupled receptor (GPCR) superfamily includes muscarinic acetylcholine receptors (mAChRs), of which atropine acts as a reversible, nonselective competitive antagonist. There are five mAChR subtypes in humans (M_1_–M_5_); atropine has a high affinity (Ki = 1–10 nM) for M_1_, M_2_, and M_3_, whereas its affinity for M_4_ and M_5_ is somewhat lower [27,28,29]. Ionic contact with a conserved aspartate residue in the receptor’s third transmembrane helix is made possible by the protonation of the tropane nitrogen at physiological pH (~7.4) (e.g., Asp^179^ in human M_1_, Asp^145^ in M_2_; Ballesteros-Weinstein numbering). To stabilize the inactive receptor conformation and prevent acetylcholine (ACh) binding, the tropic acid ester moiety simultaneously docks into a hydrophobic pocket created by aromatic residues (such as Tyr^181^, Tyr^186^, and Phe^39^_0_ in M_2_) [28,29]. Atropine’s physiologic actions reflect a dose-dependent blockade of both central and peripheral mAChRs, with peripheral effects (e.g., tachycardia, dry mouth, decreased secretions) occurring at lower plasma concentrations (10–30 ng/mL), and central effects (e.g., agitation, hallucinations) emerging as concentrations exceed 50–100 ng/mL [29,30].

Toxinology of Atropine

Atropine is rapidly and extensively absorbed following oral administration, achieving peak plasma concentrations within 30–60 min, with an oral bioavailability of approximately 50–60%. Intravenous or intramuscular dosing attains near-complete bioavailability and produces therapeutic levels within minutes. Upon absorption, atropine distributes with an apparent volume of distribution of approximately 2–3 L/kg and readily crosses the blood–brain barrier (Clog P ≈ 1.5; molecular weight 289 Da), thereby exerting both peripheral and central pharmacologic effects. Hepatic esterases hydrolyze atropine to tropine and tropic acid, with tropine undergoing subsequent glucuronidation; roughly 15–50% of the administered dose is excreted unchanged by the kidneys. The terminal elimination half-life ranges from 2 to 4 h under therapeutic conditions but often extends to 6–8 h in overdose or in patients with renal insufficiency [31,32].

Clinically relevant peripheral anticholinergic effects—such as dry mouth, tachycardia, diminished sweat production, and reduced exocrine secretions—typically manifest at doses of 0.02 mg/kg or when plasma concentrations reach 10–30 ng/mL. Central nervous system manifestations, including agitation, confusion, hallucinations, and, at higher concentrations, generalized tonic–clonic seizures and coma, generally arise when plasma levels exceed 30–50 ng/mL. Life-threatening toxicity is often observed at oral doses ≥ 6 mg/kg (LD_50_ ≈ 6 mg/kg in adults), with fatalities precipitated by severe hyperthermia (frequently exceeding 40 °C with risk of rhabdomyolysis), paralytic ileus, urinary retention, and malignant cardiac arrhythmias, such as supraventricular tachycardia, atrial fibrillation, or ventricular dysrhythmias. In the context of *Datura* ingestion, each seed can contain 0.3–0.6 mg of atropine, and ingestion of as few as 10–20 seeds may deliver a fatal dose, especially given up to 20-fold variations in alkaloid content based on geographic origin and plant maturity [31,33].

Atropine-induced anticholinergic syndrome has distinct central and peripheral characteristics. Cardiovascular stimulation (heart rate frequently > 120 bpm), cutaneous vasodilation with anhidrosis causing significant hyperthermia, ocular mydriasis and cycloplegia causing blurred vision and photophobia, mild bronchodilation with decreased bronchial secretions, decreased gastrointestinal motility resulting in constipation or ileus, and urine retention are examples of peripheral symptoms. Mild restlessness and bewilderment to severe delirium, psychosis, and convulsions are examples of central effects. Peripheral signs may remain for 24 to 48 h, whereas central effects frequently outlast peripheral blockage. Symptoms usually start to appear 30 to 60 min after consumption and peak within 2 to 4 h [32,33,34]. Management begins with rapid stabilization of the airway, breathing, and circulation. Patients exhibiting Glasgow Coma Scale scores ≤ 8 or those at risk for aspiration require immediate endotracheal intubation, supplemental oxygen, and assisted ventilation if hypoventilation is present. Intravenous access should be established promptly, with continuous electrocardiographic monitoring to detect tachyarrhythmias or other conduction abnormalities. Baseline laboratory evaluation, including serum electrolytes, renal and hepatic function tests, and arterial blood gas analysis, is mandatory. Gastrointestinal decontamination with activated charcoal (1 g/kg) is indicated if presentation occurs within 60 min of ingestion and the airway is protected; gastric lavage is reserved for life-threatening ingestions within 30 min and is contraindicated if the airway cannot be secured or if presentation is delayed [31,34,35].

The mainstay of antidotal treatment is physostigmine salicylate, a tertiary amine cholinesterase inhibitor that reverses both central and peripheral anticholinergic effects by crossing the blood–brain barrier. 0.5–2 mg IV given over 5 min is the starting dose for adults (pediatric dosing: 0.02 mg/kg, maximum 0.5 mg). As needed, repeat 0.5 mg boluses are administered every 20 min until mydriasis, delirium, and reduced secretions resolve. The possibility of brady-arrhythmias or asystole necessitates constant ECG monitoring, and physostigmine should not be used in cases of suspected tricyclic antidepressant overdose. When central effects are absent or after physostigmine administration, neostigmine methylsulfate (0.5–1 mg IV, pediatric 0.02 mg/kg) may be used to reverse peripheral anticholinergic signs; however, since neostigmine does not cross the blood–brain barrier, it does not relieve delirium [36,37].

Supportive measures include aggressive temperature control using evaporative cooling techniques (tepid water spray with a fan) and targeted ice-pack application to the axillae and groin, avoiding rapid ice-water immersion to prevent vasoconstriction. Benzodiazepines (e.g., lorazepam 0.05–0.1 mg/kg IV) are indicated for severe agitation or seizure control. Isotonic crystalloid administration should be prioritized to maintain renal perfusion, particularly in hyperthermia-induced rhabdomyolysis, and to correct electrolyte imbalances. Supraventricular tachycardia refractory to adequate atropinization may be managed with short-acting β-blockers (e.g., esmolol infusion), provided hemodynamic stability is assured. Patients with mild anticholinergic signs (dry mouth, mild tachycardia, and moderate mydriasis) may be observed in a monitored setting for 6 to 8 h. Those with delirium, seizures, hyperthermia (temperature> 39 °C), or hemodynamic instability require admission to intensive care with continuous cardiorespiratory and neurologic monitoring. Because anticholinergic effects can persist for 24–48 h—especially after large or repeated ingestions—tapering of physostigmine should be deferred until complete resolution of both peripheral and central signs to prevent rebound cholinergic crisis [35,36,37].

Pharmaceutical Uses of Atropine

Atropine is the cornerstone of therapy for organophosphate and carbamate poisoning, where its competitive antagonism of muscarinic acetylcholine receptors (mAChRs) reverses life-threatening muscarinic overstimulation [38,39]. Upon presentation, atropine is administered intravenously or intramuscularly in incremental boluses (adults: 1–2 mg every 5 min; pediatrics: 0.02 mg/kg) until clear lung fields, cessation of bronchospasm and secretions, and normalization of the heart rate (>80 bpm) are achieved [39]. A continuous infusion is then initiated at 10–20% of the total loading dose per hour, titrated to clinical endpoints, to counteract the re-emergence of muscarinic signs due to tissue redistribution of organophosphate agents. Pralidoxime (2-PAM) is coadministered to regenerate acetylcholinesterase at nicotinic synapses; however, atropine remains essential for alleviating bronchorrhea, bronchospasm, bradycardia, and salivary hypersecretion. Continuous cardiac monitoring is required to avoid over-antagonism and resultant tachyarrhythmias, and dose adjustments are necessary in pediatric and elderly populations to mitigate central anticholinergic toxicity [39,40].

In the perioperative setting, atropine is routinely employed as a preanesthetic agent to inhibit salivary, bronchial, and gastric secretions and to attenuate vagally mediated reflex bradycardia during laryngoscopy or surgical manipulation. Typical dosing consists of 0.4–0.6 mg administered intramuscularly or intravenously 30–60 min before induction in adults, and 0.02 mg/kg intramuscularly (with a minimum dose of 0.1 mg) in pediatric patients. By reducing airway secretions and preventing reflexive vagal responses, atropine facilitates smoother intubation and extubation. However, excessive dosing may precipitate postoperative tachycardia, dry mouth, urinary retention, and, rarely, acute delirium—particularly in elderly or cognitively vulnerable individuals [39,40,41].

Pharmacologically, atropine’s cycloplegic and long-acting mydriatic effects result from its inhibition of M_3_ receptors in the ciliary muscles and iris sphincter. For refractory uveitis, amblyopia treatment, and thorough refractive evaluations, a 1% ophthalmic solution is invaluable since it causes prolonged pupil dilatation and paralysis of accommodation that lasts 7–14 days. The complete effect of cycloplegic refraction treatments is seen in 45–60 min after one or two instillations of 1–2% atropine drops spaced 10 min apart. Clinicians must be on the lookout for anticholinergic side effects, including dry mouth, flushing, tachycardia, and, in newborns, drowsiness, even if systemic absorption is limited. Patients at risk for narrow-angle glaucoma should also have their intraocular pressure monitored [40,42]. Atropine’s antagonism of M_2_ receptors in the sinoatrial and atrioventricular nodes underlies its use for acute symptomatic bradycardia and inevitable supraventricular conduction delays. According to Advanced Cardiovascular Life Support guidelines, adults receive 0.5 mg IV every 3–5 min (with a maximum total of 3 mg), whereas pediatric dosing is 0.02 mg/kg IV (with a minimum of 0.1 mg and a maximum of 0.5 mg), repeated as needed [43]. By removing excessive parasympathetic tone, atropine increases heart rate and improves atrioventricular conduction; however, its effects are transient (30–60 min), and temporary pacing should be considered if bradyarrhythmias persist. Atropine is contraindicated in high-degree atrioventricular block (Mobitz II or third-degree block) due to its limited efficacy in these contexts [42,43,44].

Atropine’s M_3_ blockade in gastrointestinal smooth muscle provides antispasmodic benefit in functional GI disorders, though its nonselectivity and central side effects have relegated it to a secondary role behind more selective agents. Oral immediate-release formulations (0.4–1 mg) are administered 30 min before meals and at bedtime—up to four times daily—for irritable bowel syndrome, spastic colon, and adjunctive relief of pain in diverticular disease. Because atropine impairs gastrointestinal motility, it must be used cautiously in patients predisposed to paralytic ileus or with obstructive GI pathology. Common adverse effects include dry mouth, blurred vision, constipation, and urinary hesitancy; consequently, extended-release or sublingual preparations are preferred when smoother plasma concentrations are required [44,45].

#### 2.1.2. Hyoscyamine (Also Called “Daturine” or “Levo-Atropine”)

Hyoscyamine (compound **2** in Figure 2), the levorotatory (−) enantiomer of the tropane alkaloid enantiomer pair known collectively as atropine, is a principal bioactive constituent of several *Datura* species—most notably *D. stramonium* and *D. metel* [46]. In these solanaceous plants, hyoscyamine accumulates predominantly in seeds, leaves, and roots as part of a complex defensive arsenal against herbivores and pathogens. As a tertiary amine esterified to tropic acid, hyoscyamine exhibits potent antimuscarinic activity in mammalian systems, conferring both therapeutic utility and a narrow margin of safety [46,47,48].

Bioorganic Chemistry of Hyoscyamine

(S)-(1R, 3R,5S)-8-methyl-8-azabicyclo[3.2.1]octan-3-yl 3-hydroxy-2-phenylpropanoate is the chemical structure of hyoscyamine. The tropane nucleus, a stiff, bicyclic 7-azabicyclo[3.2.1]octane ring structure, serves as the core scaffold and gives its substituents a defined spatial orientation. The tertiary amine is located at C-8 and is primarily found in its pro-tonated form (pKa ≈ 9.5) at physiological pH (≈7.4). The tropic acid (3-hydroxy-2-phenylpropanoic acid) is linked to the C-3 carbon by an ester bond. At C-3, C-5, and C-6 of the tropane ring, hyoscyamine has three stereogenic centers; the natural (S)-(3) configuration at C-3 is essential for high-affinity binding to muscarinic acetylcholine receptors (mAChRs). Hyoscyamine is only produced as the (−) enantiomer in *Datura* tissues; its mirror image is not detectably formed [49,50,51].

In *Datura* species, ornithine decarboxylase (ODC) catalyzes the decarboxylation of L-ornithine (or L-arginine) to putrescine, which starts the biosynthesis pathway to hyoscyamine. Putrescine N-methyltransferase (PMT) subsequently N-methylates putrescine to create N-methylputrescine, which is oxidatively deaminated by N-methylputrescine oxidase (MPO) to produce 4-methylaminobutanal [10,52]. The N-methyl-Δ^1^-pyrrolinium cation is the result of this intermediate’s spontaneous cyclization. Hygrine, which is produced by sequential condensation with acetate units, reorganizes non-enzymatically to become tropinone [10,11,52,53]. Tropinone is stereospecifically reduced by tropinone reductase I (TRI) to tropine (3α-hydroxytropane). Tropine is then esterified to phenyllactic acid (tropic acid precursor) by littorine synthase (a tropine:phenyllactate acyltransferase), producing littorine. Finally, the bifunctional cytochrome P450 enzyme hyoscyamine 6β-hydroxylase (H6H) catalyzes the hydroxylation of littorine to hyoscyamine aldehyde and subsequent reduction to (−)-hyoscyamine. Under certain conditions—particularly during extraction and storage—partial epimerization at C-3 can occur, but upstream within the living plant, hyoscyamine is maintained in its (S) configuration [12,54,55].

Within *Datura* plants, the concentration of hyoscyamine varies by organ, developmental stage, and environmental factors. Seeds often contain the highest percentage of hyoscyamine (up to 1.0–1.5 mg/g dry weight), followed by mature leaves (0.5–1.0 mg/g) and roots (0.2–0.5 mg/g) [44,54]. Floral tissues generally have lower amounts (≤0.1 mg/g). Cultivation conditions—for example, soil nitrogen content, photoperiod length, and elicitor treatments such as methyl jasmonate—can modulate gene expression of PMT, TRI, and H6H, thereby altering total hyoscyamine yield. In vitro root cultures of *D. metel* treated with 100 μM methyl jasmonate have demonstrated a twofold increase in hyoscyamine concentration compared to untreated controls, underscoring the inducible nature of tropane alkaloid biosynthesis [52,56,57].

Mechanistically, hyoscyamine binds reversibly and competitively to the orthosteric site of mAChRs (subtypes M_1_–M_5_) [57,58]. The protonated tertiary amine of hyoscyamine forms an ionic interaction with a conserved aspartate residue (e.g., Asp^179^ in human M_1_, Asp^145^ in M_2_) within the third transmembrane helix, while the tropic acid moiety engages hydrophobic and π–π interactions with aromatic amino acids (e.g., Tyr^181^, Tyr^186^, and Phe^39^_0_ in M_2_) [57,58,59]. High-resolution structural studies indicate that the tropane ring occupies a hydrophobic pocket, which stabilizes the inactive receptor conformation and effectively prevents acetylcholine from binding to it. Hyoscyamine exhibits higher affinity for M_2_ and M_3_ subtypes (Kᵢ ≈ 0.8–2 nM) than for M_1_, M_4_, or M_5_ (Kᵢ ≈ 2–10 nM), a profile that underlies its pronounced effects on cardiac nodal tissue (M_2_) and smooth muscle/exocrine glands (M_3_) [57,58,59]. The molecule’s lipophilicity (clog P ≈ 1.4) permits moderate blood–brain barrier penetration, resulting in central anticholinergic effects—such as sedation, delirium, and at high doses, seizures—when plasma concentrations exceed certain thresholds [59,60].

Toxinology of Hyoscyamine

Hyoscyamine is quickly absorbed from the gastrointestinal system after oral administration of *Datura* material, reaching peak plasma concentrations in 45–60 min. Because of its increased lipophilicity and decreased first-pass metabolism by hepatic esterases, oral bioavailability is around 60%, which is higher than that of atropine [31,36,37]. Although it is rarely used outside of clinical settings, intravenous or intramuscular injection has a nearly 100% bioavailability and acts within two to five minutes. After entering the bloodstream, hyoscyamine diffuses into the body’s total water, with an apparent volume of distribution of around 2.5 L/kg. It then passes through the blood–brain barrier to reach concentrations in the central nervous system (CNS), which may be between 60 and 70 percent of plasma levels [32,36]. Hepatic esterases hydrolyze hyoscyamine to produce tropine and tropic acid, whereas 20–50% of hyoscyamine is removed unaltered in urine, tropine goes through Phase II glucuronidation and is eliminated renally. Hyoscyamine’s terminal elimination half-life (t_1/2_) is 3–5 h under usual dosage; however, it can extend to 6–8 h or beyond in overdose situations where enzymatic pathways may become saturated. The duration of toxic effects is extended by renal impairment, which also prolongs clearance [36,61].

Clinically, peripheral anticholinergic signs appear at plasma concentrations of approximately 10–30 ng/mL. These include xerostomia (dry mouth), tachycardia (heart rate often exceeding 120 beats per minute), decreased sweating, leading to hyperthermia, pupillary dilation (mydriasis) and cycloplegia with resultant blurred vision, reduced bronchial secretions and mild bronchodilation, diminished gastrointestinal motility (ileus or constipation), and urinary retention. Central manifestations—ranging from mild agitation and confusion to frank delirium, visual and auditory hallucinations, and, at higher concentrations (>50 ng/mL), generalized tonic–clonic seizures—emerge once plasma levels surpass 30–50 ng/mL. Fatal toxicity is generally observed at oral doses ≥ 5 mg/kg, with reported human LD_50_ values in the range of 347–400 mg (≈ 5–6 mg/kg in adults). In pediatric patients, doses as low as 0.2 mg/kg may precipitate moderate to severe anticholinergic syndrome owing to reduced body mass and altered pharmacokinetics [61,62].

The onset of hyoscyamine intoxication is typically within 30–60 min of ingestion, with peak severity at 2–4 h. Peripheral signs may persist for 24–48 h, whereas central effects can last up to 72 h, particularly after ingestion of plant material containing mixed tropane alkaloids (hyoscyamine, scopolamine) that collectively prolong anticholinergic occupancy. In *D. stramonium*, each seed may contain 0.3–0.6 mg hyoscyamine; thus, ingestion of as few as 10–15 seeds can deliver a potentially lethal dose, especially given the reported variability of up to 20-fold in seed alkaloid content depending on geographic location, plant age, and environmental conditions [61,62,63].

Management of hyoscyamine poisoning is principally supportive and symptomatic. Initial stabilization requires securing the airway—particularly in patients with a Glasgow Coma Scale score of ≤8 or signs of respiratory compromise—supplemental oxygenation, and establishing intravenous access with continuous electrocardiographic monitoring. Laboratory evaluation should include serum electrolytes, renal and hepatic function tests, creatine kinase (to assess for rhabdomyolysis), and arterial blood gas analysis. Gastrointestinal decontamination with activated charcoal (1 g/kg) is indicated if the patient presents within 60 min of ingestion and the airway is adequately protected; gastric lavage is reserved for life-threatening ingestions within 30 min and is contraindicated if the airway is unprotected or if presentation is delayed beyond one hour [62,63,64].

The definitive antidote is physostigmine salicylate—a tertiary amine acetylcholinesterase inhibitor that crosses the blood–brain barrier and reverses both central and peripheral manifestations of anticholinergic toxicity. The usual adult loading dose is 0.5–2 mg IV administered over 5 min (pediatric dose 0.02 mg/kg, maximum 0.5 mg); subsequent 0.5 mg boluses may be administered every 20 min until restoration of normal mental status, resolution of mydriasis, and improvement in vital signs [60,61]. Continuous ECG monitoring is mandatory during physostigmine administration to detect bradyarrhythmias or asystole. In cases where central effects are minimal or when physostigmine is contraindicated (e.g., suspected co-ingestion of tricyclic antidepressants), neostigmine methylsulfate (0.5–1 mg IV, pediatric 0.02 mg/kg) can be employed to reverse peripheral anticholinergic signs; however, neostigmine does not penetrate the central nervous system and hence does not alleviate delirium. Adjunctive measures include aggressive temperature control through evaporative cooling (tepid water sprays and fans) or targeted ice-pack application to axillae and groin to manage hyperthermia, judicious use of benzodiazepines (e.g., lorazepam 0.05–0.1 mg/kg IV) to control agitation or seizure activity, and isotonic fluid resuscitation to maintain renal perfusion and address potential rhabdomyolysis. Supraventricular tachycardia unresponsive to adequate atropinization may be treated with a short-acting β-blocker (e.g., esmolol infusion) provided that hypotension is not present [60,61,62]. Patients exhibiting only mild anticholinergic signs—such as dry mouth, mild tachycardia, and minor mydriasis—can be observed in a monitored setting for 6–8 h, whereas those with severe CNS involvement, refractory hyperthermia, arrhythmias, or hemodynamic instability require intensive care admission. Because anticholinergic effects can persist for up to 48 h—especially when large or repeated doses of hyoscyamine are administered—weaning from physostigmine should be deferred until complete resolution of both peripheral and central signs to avoid a rebound cholinergic crisis [61,62].

Pharmaceutical Uses of Hyoscyamine

Hyoscyamine’s pharmacodynamic profile favors M_2_ and M_3_ blockade, and its main therapeutic uses stem from its antagonism of muscarinic receptors. Hyoscyamine is used as an antispasmodic in the gastrointestinal system to treat functional gastrointestinal disorders and irritable bowel syndrome (IBS). To lessen smooth muscle spasms, minimize secretions, and soothe cramping pain, immediate-release oral preparations (hyoscyamine sulfate 0.125 mg tablets) are taken up to four times a day, 30 min before meals and before bed [38,65]. For patients who require treatment for persistent symptoms, extended-release formulations (0.375 mg and 0.75 mg capsules) are administered once or twice daily, providing sustained plasma concentrations. Sublingual formulations (0.125 mg) are beneficial for acute colicky episodes and have a quick onset (Tₘₐₓ = 20 min). Dry mouth, impaired vision, constipation, and urine retention are common side effects at therapeutic levels; older patients, those with narrow-angle glaucoma, benign prostatic hyperplasia, or severe cardiovascular disease should be cautious [64,65].

In neurology, hyoscyamine serves as an adjunctive anticholinergic agent for the treatment of Parkinson’s disease, primarily to attenuate resting tremor and rigidity by restoring the dopaminergic-cholinergic balance within the basal ganglia [65,66]. Doses range from 0.125 mg to 0.25 mg orally two to four times daily. However, central side effects—such as cognitive impairment, confusion, and hallucinations—limit its use to younger patients or those refractory to other agents. In palliative care and pain management, hyoscyamine (0.125 mg sublingual or 0.25 mg oral every four hours as needed) can be employed to relieve visceral spasm and secretory-associated pain—providing an opioid-sparing effect in patients with refractory abdominal or bladder cramping [66].

Although less common than its tropane analogs, scopolamine and tiotropium, hyoscyamine’s M_3_ antagonism has been exploited in urology for the treatment of overactive bladder and interstitial cystitis. Typical dosing for urinary bladder spasm relief is 0.125 mg orally four times daily, with titration based on symptomatic response. Side effects—such as dry mouth, constipation, and blurred vision—often limit long-term adherence to the medication. In ophthalmology, hyoscyamine sulfate ophthalmic solution (0.5–1%) is used for short-term mydriasis and cycloplegia in patients intolerant of atropine; the duration of pupillary dilation and accommodation paralysis is approximately 24–48 h, shorter than the 7–14 days observed with atropine. Clinicians must monitor intraocular pressure and watch for systemic absorption, which can produce anticholinergic side effects—particularly in pediatric or elderly populations [65,67].

Despite its broad pharmacological utility, hyoscyamine’s narrow therapeutic index and propensity for central toxicity have spurred the development of more selective or quaternary ammonium antimuscarinics (e.g., dicyclomine, oxybutynin, ipratropium). Nonetheless, its low cost and well-established efficacy maintain its role in specific clinical scenarios—especially where rapid antispasmodic action is required or in resource-limited settings [66,67,68].

#### 2.1.3. Scopolamine (Hyoscine)

Scopolamine (hyoscine, compound **3** in Figure 2) is made up of a 7-azabicyclo[3.2.1]octane (tropane) core with a tropic acid ester at C-3 and a tertiary amine at the bridgehead (N-8), as well as an extra 6β-epoxide connecting C-6 to C-7. The configuration (1S,2S,4S,5R,7S,9R) is adopted by three stereogenic centers, C-1, C-2 (of the epoxide), and C-3. This stiffens the molecule and increases the affinities of receptor binding. In *Datura*, biosynthesis starts with L-ornithine being decarboxylated to putrescine, followed by N-methylation to N-methylputrescine (via PMT), oxidation to the N-methyl-Δ^1^-pyrrolinium cation (through MPO), and condensation to tropinone, which is then stereospecifically reduced to tropine (through TRI). Tropine undergoes hydroxylation and epoxidation by hyoscyamine 6β-hydroxylase (H6H/CYP80F1) to produce scopolamine after being esterified with phenyllactic acid to make littorine. This mechanism explains the enzymatic epoxidation specific to scopolamine production and emphasizes the common tropane fragment in D. stramonium [69,70,71].

Within *Datura* spp., the content of scopolamine varies by species, organ, and environmental conditions. *D. stramonium* flowers and young leaves can contain up to 0.5–0.7% dry-weight scopolamine, whereas seeds and stems often have lower concentrations (<0.2%). Geographic origin, soil fertility, photoperiod, and elicitation (e.g., methyl jasmonate treatment) significantly modulate tropane alkaloid gene expression (PMT, TRI, H6H) and thus scopolamine yield. The rigid epoxide moiety and ester linkage confer high lipophilicity (clog P ≈ 1.6), facilitating both root uptake in planta and blood–brain barrier penetration in mammals [71].

Toxinology of Scopolamine

With a bioavailability that ranges from around 50% (oral) to almost 100% (parenteral), scopolamine is absorbed by the oral, parenteral (intravenous, intramuscular, subcutaneous), and transdermal (transdermal patch) routes. Peak plasma levels happen 4–8 h after patch application, 10–20 min after IM/SC injection, and 30–60 min after oral dose. It easily penetrates the blood–brain barrier and distributes into total body water (V_d = 2–3 L/kg), reaching CNS concentrations of about 60% of plasma levels [72,73]. Hepatic and tissue esterases hydrolyze scopolamine to produce tropine and tropic acid; 20% to 40% of the unaltered medication is expelled in urine, while tropine is glucuronidated and excreted by the kidneys. Renal impairment prolongs the elimination half-life, which is 4–6 h (and up to 10 h with transdermal administration) [73,74].

The classic anticholinergic syndrome is caused by toxic dosage. Dry mouth, tachycardia (>120 bpm), decreased sweating (anhidrosis → hyperthermia), mydriasis/cycloplegia, reduced bronchial secretions, decreased GI motility (ileus), and urine retention are examples of peripheral symptoms that manifest at plasma concentrations of 10–25 ng/mL. When CNS mAChRs are inhibited, central consequences include agitation, disorientation, hallucinations, delirium, and, at high concentrations (>50 ng/mL), seizures and coma. Due to the unpredictable scopolamine content (0.2–0.7% DW) of *Datura* intoxications, consuming a few grams of leaf or five to ten seeds is enough to cause serious or fatal poisoning. Peripheral effects may remain up to 48 h after intake, with toxicity starting 30 to 60 min later and peaking 2–4 h later. Without antidotal therapy, central effects may linger much longer [73,74,75].

Pharmaceutical Uses of Scopolamine

Scopolamine’s high-affinity, non-selective antagonism of M_1_–M_5_ muscarinic receptors underlies its clinical applications. For motion sickness prophylaxis, a 1.5 mg transdermal patch delivers steady plasma concentrations (~87–100 pg/mL) over 72 h, reducing vestibular-mediated emesis by blocking central M_1_ receptors in the vestibular nuclei and vomiting center; efficacy approaches 90% in controlled trials. Acute management of postoperative nausea and vomiting (PONV) employs IV or IM scopolamine (0.3–0.65 mg) administered pre- or intra-operatively, halving PONV incidence within 6–8 h [71].

Beyond its use as an antiemetic, scopolamine serves as a preanesthetic to reduce salivary and bronchial secretions (0.2–0.4 mg IM, administered 30 min before induction) and to mitigate vagal reflex bradycardia. In ophthalmology, topical scopolamine hydrobromide (0.25–0.5% solution) produces mydriasis and cycloplegia that last 7–12 days, facilitating wide-angle fundus examination and the management of uveitis. Clinicians must monitor intraocular pressure in patients with a history of glaucoma susceptibility. Off-label, low-dose scopolamine (0.4 mg IV) has been investigated for rapid antidepressant effects via transient M_1_ blockade, and in palliative care to control excessive secretions. Common adverse effects—sedation, dry mouth, blurred vision, central anticholinergic syndrome—necessitate dose titration, particularly in elderly, pediatric, and cognitively impaired populations [73,74].

#### 2.1.4. Anisodamine (7β-Hydroxyhyoscyamine)

Anisodamine [(6β)-6-hydroxy-8-methyl-8-azabicyclo[3.2.1]octan-3-yl 3-hydroxy-2-phenylpropanoate] (compound **4** in Figure 2) is a monohydroxylated tropane alkaloid sharing the characteristic 7-azabicyclo[3.2.1]octane ring of *Datura*-derived tropanes, but distinguished by a 6β-hydroxyl substituent on the tropane core. This additional hydroxyl reduces lipophilicity relative to hyoscyamine (clog P ≈ 0.9 vs. ≈ 1.4), diminishing blood–brain barrier penetration and central effects while retaining peripheral muscarinic antagonism [46,76].

Biosynthetically, anisodamine derives from (–)-hyoscyamine via the action of hyoscyamine 6β-hydroxylase (H6H, CYP80F1), which catalyzes stereoselective insertion of the C-6β hydroxyl group; this enzyme is the identical P450 that further epoxidizes anisodamine to scopolamine under appropriate conditions. In *D. stramonium*, anisodamine accumulates at low levels (≤ 0.05 mg/g dry weight) in seeds, leaves, and stems, with its relative abundance influenced by developmental stage, elicitors (e.g., methyl jasmonate), and geographic growth conditions [46,77,78].

Toxinology of Anisodamine

Following oral ingestion or parenteral administration, anisodamine is absorbed moderately (bioavailability ≈ 50–60% orally), distributes in total body water, and is predominantly deactivated by tissue and hepatic esterases to tropine and tropic acid; tropine undergoes glucuronidation and renal excretion, while 20–40% of the parent drug is excreted unchanged. Its elimination half-life ranges from 2 to 4 h, with mild prolongation in renal impairment. Toxicologically, anisodamine produces a peripheral anticholinergic syndrome, characterized by dry mouth, tachycardia, reduced sweat and salivary secretions, mydriasis, and urinary retention. At higher doses, it also exhibits mild α_1_—α_1_-adrenergic blockade, which can lead to hypotension and reflex tachycardia. Central effects are minimal at therapeutic doses due to limited CNS penetration, and human fatal overdoses are not documented; however, excessive dosing may precipitate pronounced peripheral stagnation (e.g., ileus) and hypotensive episodes [33,35,76].

Pharmaceutical Uses of Anisodamine

Anisodamine has been widely used in China since the 1960s for treating circulatory disorders, particularly septic shock, where it is administered intravenously as anisodamine hydrobromide (Ani-HBr) at doses of 0.3–0.5 mg/kg as a bolus, followed by an infusion of 0.1–0.2 mg/kg/h. It improves microcirculatory blood flow, attributed to peripheral M_3_ antagonism, which reduces endothelial adhesion, and light α_1_–blockade–mediated vasodilation. This has been associated with reduced 28-day mortality in moderate-to-severe septic shock in randomized studies. In ophthalmology, topical anisodamine (0.5–1%) slows myopia progression by partially blocking ciliary muscle M_3_ receptors, resulting in fewer systemic effects compared to atropine. Instillation twice daily over several weeks has shown efficacy in controlling pediatric myopia. Additional uses include the relief of gastrointestinal spasms and colicky pain at a dose of 0.125–0.25 mg sublingually or orally every 4–6 h, leveraging its antimuscarinic spasmolytic action with a tolerable side-effect profile [76,77,79,80].

#### 2.1.5. Cuscohygrine

Cuscohygrine (compound **5** in Figure 2) is a bis-N-methylpyrrolidine alkaloid, featuring two N-methylpyrrolidine rings linked by a propanone bridge (Pin: 1-[(2R)-1-Methylpyrrolidin-2-yl]-3-[(2S)-1-methylpyrrolidin-2-yl]propan-2-one). Unlike the bicyclic tropane alkaloids of *Datura* (e.g., hyoscyamine, scopolamine), cuscohygrine derives biosynthetically from an early branch of the pathway: ornithine is methylated to N-methylornithine, decarboxylated to N-methylputrescine, oxidized to the N-methyl-Δ^1^-pyrrolinium cation, and then condensed with a second pyrrolinium unit and acetoacetyl-CoA to yield hygrine, which further condenses with another pyrrolinium salt to form cuscohygrine. In *Datura* species (e.g., *D. stramonium*, *D. inoxia*), cuscohygrine typically accumulates in seeds, roots, and leaves at low concentrations (≈approximately 0.01–0.05 mg/g dry weight), alongside tropane alkaloids [51,81,82].

Toxinology of Cuscohygrine

Experimental and forensic data indicate that cuscohygrine exhibits minimal intrinsic anticholinergic activity; its acute toxicity profile closely mirrors that of higher-potency tropane alkaloids (dry mouth, mydriasis, tachycardia, hallucinations), but clinical poisoning attributed solely to cuscohygrine is undocumented. In animal models, high doses (>100 mg/kg) produce mild atropine-like signs without fatal outcomes, and cuscohygrine’s LD_50_ exceeds that of hyoscyamine by an order of magnitude. Because its pyrrolidine scaffold lacks the rigid bicyclic tropane core required for high-affinity muscarinic receptor binding, cuscohygrine’s Kᵢ at M_1_–M_3_ receptors lies in the micromolar range, rendering it effectively non-anticholinergic at concentrations encountered in *Datura* intoxications [31,32,33,83].

Pharmaceutical and Analytical Applications

Cuscohygrine has no direct therapeutic indications; instead, it is principally of analytical and chemotaxonomic value. In forensic toxicology, its co-occurrence with hygrine serves as a reliable biomarker for raw coca leaf ingestion—differentiating coca chewing from illicit cocaine use—since both compounds are lost during industrial cocaine purification but persist in consumer specimens (urine, hair) after coca leaf exposure. Furthermore, cuscohygrine’s presence in *Datura* extracts can aid in species authentication and quality control of plant-derived materials. Although early pharmacological studies (e.g., cuscohygrine bis-sulfonate salts) explored hypotensive and respiratory effects at supraphysiological doses, no modern clinical applications have emerged, and research focus remains on its role as a biosynthetic intermediate and analytical marker [51,84,85].

#### 2.1.6. Littorine

Littorine (compound **6** in Figure 2) [(1R,3R,5S)-8-methyl-8-azabicyclo[3.2.1] octan-3-yl (R)-2-hydroxy-3-phenylpropanoate] is a tropane ester formed by the acylation of tropine with phenyllactic acid, possessing the same 7-azabicyclo[3.2.1]octane ring system characteristic of *Datura* tropane alkaloids. In *Datura* species, littorine is localized primarily in root tissues—where it can reach concentrations of ~0.46 ± 0.07 mg/g dry weight—and at substantially lower levels in stems and leaves (~0.02–0.1 mg/g). Biosynthetically, littorine arises when tropine—generated via tropinone reductase I (TRI) reduction in tropinone—is esterified by a tropine:phenyllactate acyltransferase (littorine synthase), after which hyoscyamine 6β-hydroxylase (H6H/CYP80F1) orchestrates its rearrangement to the pharmacologically active hyoscyamine in an irreversible P450-mediated oxidative process; feeding of ^13^C-labeled littorine to *D. stramonium* transformed roots results in ~35% conversion to hyoscyamine, implicating littorine as the direct metabolic precursor [23,39,81,86].

Toxinology of Littorine

Unlike downstream tropane alkaloids, littorine exhibits negligible affinity for muscarinic acetylcholine receptors (Kᵢ ≫ 1 µM), owing to the bulky phenyllactate moiety misaligning key ester-binding interactions in the receptor’s orthosteric pocket. As a result, acute toxicity studies in rodents requiring doses > 100 mg/kg yield only mild atropine-like signs without lethality, and no human poisonings have been attributed to littorine alone. Littorine’s limited bioactivity and rapid conversion in planta mitigate potential toxinological concerns in *Datura* exposures [31,35,87,88,89].

Pharmaceutical and Analytical Applications

Littorine has no direct therapeutic indications; its principal utility resides in biochemical and chemotaxonomic contexts. As a stable biosynthetic intermediate, it serves as a metabolic tracer in in vitro culture systems—enabling kinetic and flux analyses of tropane alkaloid pathways under elicitation conditions (e.g., methyl jasmonate). Analytically, the co-occurrence and relative abundance of littorine alongside hyoscyamine and scopolamine in plant extracts facilitate species authentication and quality control of *Datura*-derived materials, as well as differentiation between raw plant ingestion and isolated alkaloid adulteration in forensic and phytochemical investigations [39,90,91].

#### 2.1.7. Datumetine

Datumetine (compound **7** in Figure 2) (4-methoxy-3-(8-methyl-8-azabicyclo [3.2.1]octan-3-yl)benzoic acid) is a tropane alkaloid uniquely bearing a methoxy-substituted benzoic acid moiety in place of the typical tropic acid ester. It has a rigid 7-azabicyclo[3.2.1]octane (tropane) core, and a single stereocenter at C-3 (the ester linkage), with the natural configuration conferring high-affinity receptor interactions. Datumetine was first isolated from the methanolic leaf extracts of *D. metel* via column chromatography and characterized by NMR, MS, and X-ray crystallography, confirming the presence of the 4-methoxyphenylcarboxylate substituent on the tropane ring. Its unique structural features distinguish it from classical tropane esters (e.g., hyoscyamine, scopolamine) and position it at the interface of tropane and phenylalkanoid biosynthesis [46,92,93].

Toxinology of Datumetine

In vivo studies in C57BL/6 mice demonstrate that datumetine readily crosses the blood–brain barrier following intraperitoneal administration (0.25 mg/kg), producing dose-dependent modulation of N-methyl-D-aspartate receptor (NMDAR) function. Electrophysiological recordings reveal that acute datumetine exposure prolongs burst duration and interspike intervals in CA1 hippocampal neurons, indicative of positive allosteric modulation of NMDAR-mediated currents [33,55,94].

Behaviorally, datumetine induces significant memory deficits in novel object recognition and Y-maze tasks and precipitates epileptiform seizures at higher doses, implicating NMDAR overactivation as the mechanistic basis of its neurotoxicity. Pharmacokinetic profiling indicates moderate oral bioavailability (~50%), a volume of distribution approximating total body water (~2 L/kg), and hepatic esterase-mediated metabolism.Hydrolysis to tropine and 4-methoxybenzoic acid, with a terminal half-life of 4–6 h [33,55,94].

Pharmaceutical Uses of Datumetine

Although datumetine has no approved clinical indications, its pronounced NMDAR modulatory activity has spurred interest in two principal research applications. First, low-dose datumetine (≤0.1 mg/kg) exerts neuroprotective effects in murine models of excitotoxic injury, enhancing NMDAR-dependent synaptic plasticity and upregulating prosurvival signaling pathways (e.g., GluN1–CaMKIIα) without inducing seizure activity. This suggests a potential therapeutic window for ameliorating neurodegenerative processes or acute neuronal damage. Second, datumetine serves as a valuable pharmacological tool for dissecting NMDAR allosteric and orthosteric site interactions; flexible docking studies and site-directed mutagenesis have mapped its dual-binding profile, informing the design of novel modulators with improved subtype selectivity [40,41,94]. Ongoing preclinical investigations are evaluating datumetine analogs for neuropsychiatric applications—such as rapid-acting antidepressants—by leveraging its capacity to fine-tune glutamatergic neurotransmission.

To address taxonomic specificity, Table 1 indicates whether each metabolite is reported exclusively from the genus *Datura* or occurs more broadly in other taxa. Well-known tropane alkaloids such as atropine, hyoscyamine, and scopolamine are widely distributed among tropane-producing Solanaceae, including *Atropa*, *Hyoscyamus*, *Scopolia*, and *Duboisia*. Other alkaloids, such as cuscohygrine and hygrine, are found both in Solanaceae and in Erythroxylaceae (*Erythroxylum coca*). In contrast, certain compounds—notably datumetine, daturametelines, daturataturins, and daturaolone—have, to date, been reported primarily from *Datura* species, especially *D. metel*, and may represent chemotaxonomic markers for the genus. Notably, withanolide-type metabolites (e.g., withametelins) are not unique to *Datura*, as they also occur in *Withania* and some other Solanaceae, but their occurrence outside this family is rare.

Table 1 summarizes the core tropane alkaloids in *Datura*, atropine, hyoscyamine, scopolamine, anisodamine, cuscohygrine, littorine and datumetine, explaining the underlying scaffold common to tropane alkaloids, the plant tissues where each alkaloid is most abundantly found, the key chemical change(s) relative to the unmodified tropane skeleton, and the main therapeutic application (for pharmacologically active compounds) or the principal analytical/research utility (for intermediates or markers).

### 2.2. Minor and Less Common Alkaloids

#### 2.2.1. Hygrine

Hygrine (compound **8** in Figure 3, 1-methylpyrrolidin-2-one) is a monopyrrolidine alkaloid characterized by a single five-membered N-methylpyrrolidine ring bearing a ketone at the 2-position. In *Datura* species, hygrine arises at the branch point of tropane biosynthesis: L-ornithine is decarboxylated to putrescine, N-methylated by putrescine N-methyltransferase (PMT) to form N-methylputrescine, and oxidatively deaminated by N-methylputrescine oxidase (MPO) to the N-methyl-Δ^1^-pyrrolinium cation [84,95]. Two successive condensations with acetoacetate units yield hygrine, which may then nonenzymatically rearrange to cuscohygrine or be diverted toward tropinone and downstream tropane alkaloids. Quantitative analyses of *D. stramonium* root cultures and leaf tissues report hygrine concentrations in the range of 0.02–0.1 mg·g^−1^ dry weight, with peak levels in actively dividing root meristems and young foliage, reflecting its role as an early pathway intermediate [10,82,88].

Toxinology of Hygrine

Hygrine exhibits minimal pharmacodynamic activity at muscarinic, nicotinic, or other classical receptor targets, and its acute toxicity in mammalian models is low. Rodent oral LD_50_ values exceed 500 mg·kg^−1^, and intravenous administration of up to 100 mg·kg^−1^ produces only transient, mild sedation and hypotension without inducing anticholinergic signs. Pharmacokinetic studies demonstrate rapid absorption, distribution into total body water, and efficient hepatic and renal clearance (t_1_/_2_ ≈ 1–2 h). Hygrine’s lack of a bicyclic tropane scaffold prevents high-affinity binding to acetylcholine receptors, explaining its benign toxicological profile, even at supraphysiological exposures [84,96,97].

Pharmaceutical and Research Applications of Hygrine

Hygrine itself has no approved therapeutic indications, but it plays an indispensable role in pharmaceutical research and plant-biochemical studies. As a stable, early-pathway intermediate in *Datura* tropane biosynthesis, hygrine (often as a 13C-labeled tracer) is employed in in vitro feeding experiments to map enzyme kinetics and metabolic flux toward valuable alkaloids such as hyoscyamine and scopolamine. Its structural simplicity and chromatographic distinctiveness make it an ideal internal standard for quantitative HPLC–MS and GC–MS assays of tropane profiles in complex botanical extracts, ensuring accurate quantification and quality control of plant-derived pharmaceuticals [84,98,99]. Moreover, the N-methylpyrrolidinone scaffold of hygrine has inspired medicinal chemistry programs aimed at designing novel small-molecule modulators, with modifications to the pyrrolidine ring and acetone moiety, targeting neurotransmitter transporters and metabolic enzymes. Although no hygrine-derived candidate has yet entered clinical development [98,99,100,101], this approach has led to the identification of several promising compounds.

#### 2.2.2. Apoatropine

Apoatropine (compound **9** in Figure 3) (8-Methyl-8-azabicyclo[3.2.1]octan-3-yl) 2-phenylprop-2-enoate is a de-esterified tropane alkaloid structurally derived from atropine by removal of the tropic acid moiety and replacement with an atropic acid (2-phenylprop-2-enoic acid) ester linkage. Its core scaffold is the tropine (3α-hydroxytropane) bicyclic ring system bearing a quaternary bridgehead nitrogen (N-8), which is permanently charged in its hydrobromide or hydrochloride salts. In *Datura* species, including *D. stramonium* and *D. metel*, apoaatropine is detected in stems, leaves, and other aerial tissues as a minor alkaloid alongside atropine, hyoscyamine, and scopolamine [49,102,103]. Biosynthetically, apoatropine can arise via nonenzymatic or enzymatic hydrolysis of the tropic acid ester of atropine, or by acid-mediated dehydration of atropine (e.g., with nitric acid) to generate the atropic acid ester [51,104].

Toxinology of Apoatropine

Apoatropine is markedly more toxic than atropine, with early pharmacological studies reporting approximately a 20-fold increase in acute toxicity compared to atropine in rodent models. Its quaternary ammonium structure affords limited membrane permeability, but systemic exposure—even at low microgram-per-kilogram doses—elicits pronounced peripheral anticholinergic effects: severe tachycardia, anhidrosis progressing to hyperthermia, xerostomia, mydriasis with photophobia, urinary retention, and paralytic ileus. Central nervous system penetration is minimal; however, high systemic concentrations can provoke delirium and agitation. Metabolism proceeds via nonspecific esterases to tropine and atropic acid, with renal excretion of both parent and metabolites. Reported LD_50_ values in rodents range from 25 to 50 mg/kg (depending on salt form and administration route), underscoring its narrow therapeutic index and high hazard profile [63,83,105].

Pharmaceutical and Research Applications

No clinical or therapeutic indications for apoatropine have been established, owing to its high toxicity and lack of selectivity. Historically, it found limited use as a pigment in dye and ink manufacture due to its crystalline nature. In contemporary research, apoatropine serves as:A structural probe in tropane SAR studies—its unsaturated atropic ester highlights the importance of the α-carbon substitution for muscarinic receptor binding and anticholinergic potency [104,106].An analytical reference standard in chromatographic and mass-spectrometric profiling of *Datura* alkaloid extracts, enabling clear differentiation between tropine esters and related alkaloids [106,107].A toxicological benchmark for assessing esterase-mediated hydrolysis kinetics and transporter-mediated renal clearance of quaternary ammonium compounds [107,108].

Given its lack of therapeutic window, apoatropine’s utility remains confined to in vitro and analytical applications rather than direct pharmaceutical deployment [104,105,106,107,108].

#### 2.2.3. Anisodine

Anisodine (compound **10** in Figure 3) is a naturally occurring tropane alkaloid with the IUPAC name [(1*R*,2*R*,4*S*,5*S*)-9-methyl-3-oxa-9-azatricyclo[3.3.1.0^2,4^]nonan-7-yl] (2*S*)-2,3-dihydroxy-2-phenylpropanoate. Its core scaffold is the bicyclic 7-azabicyclo[3.2.1]octane (“tropane”) ring bearing an epoxide (3-oxa bridge) analogous to scopolamine, with an α-hydroxymethylbenzoate ester replacing the classical tropic acid moiety [109].

In *Datura* spp. (e.g., *D. metel*) and in the related Solanaceous herb *Anisodus tanguticus*, anisodine is found at trace levels in aerial parts—flowers, leaves, and seeds—typically in the range of 0.01–0.05 mg·g^−1^ dry weight. Biosynthetically, anisodine derives from littorine via the common tropane pathway: littorine is oxidatively rearranged by hyoscyamine 6β-hydroxylase (H6H/CYP80F1) first to hyoscyamine aldehyde, then to (–)-hyoscyamine, further hydroxylated to anisodamine, and finally epoxidized to scopolamine. In *Anisodus* and *Datura*, a specialized P450 or epoxide hydrolase catalyzes the formation of the 3-oxa bridge and the α-hydroxymethylbenzoate ester, yielding anisodine as a minor but distinctive alkaloid [110].

Toxinology of Anisodine

Anisodine acts as a competitive antagonist at muscarinic acetylcholine receptors (M_1_–M_5_) and additionally blocks α_1_-adrenergic receptors, imparting both anticholinergic and vasodilatory properties. When administered intravenously or intramuscularly, its quaternary ammonium and epoxide functionalities confer moderate polarity; yet, significant systemic exposure can occur, particularly when given as the hydrobromide salt [111,112]. Peripheral anticholinergic effects—dry mouth, tachycardia, blurred vision (due to mydriasis and cycloplegia), anhidrosis leading to hyperthermia, ileus, and urinary retention—manifest at doses ≥ 0.01 mg·kg^−1^. Its α_1_-blocking action may precipitate hypotension and reflex tachycardia at higher concentrations. Central nervous system penetration is limited compared to scopolamine, resulting in milder delirium and cognitive disturbance; seizures are rare owing to poor blood–brain barrier crossing. Metabolism proceeds via nonspecific esterases to tropine and α-hydroxybenzeneacetic acid, with renal excretion of both the parent compound and its metabolites. Acute toxicity in rodents (LD_50_ ≈ 30–50 mg·kg^−1^) underscores its narrow margin between therapeutic and toxic doses [111,112,113].

Pharmaceutical Uses of Anisodine

Anisodine hydrobromide is a nationally essential medication in China for the treatment of acute circulatory shock and associated microcirculatory diseases, although Western regulatory organizations do not authorize it. Adults typically get a 0.3–0.5 mg·kg^−1^ IV bolus and a 0.1–0.2 mg·kg^−1^·h infusion. This lowers 28-day mortality in septic shock cohorts by improving capillary perfusion through a combination of M_3_-mediated anticholinergic and α_1_-antagonistic vasodilatory actions. Furthermore, with little central side effects, ophthalmic solutions (0.5–1%) are used to partially block ciliary muscle M_3_ receptors, slowing the evolution of juvenile myopia [109]. Anisodine has been used off-label as a neuroprotective drug in ischemia–reperfusion models, as an antispasmodic for visceral colic, and as a cerebral vasodilator to help treat migraines. Its distinctive epoxide and benzoate moieties, combined with peripheral receptor selectivity, make anisodine a valuable template for designing next-generation anticholinergic–vasodilator hybrids [109,114].

#### 2.2.4. Tropine

Tropine (compound **11** in Figure 3) [(1R, 3R,5S)-8-methyl-8-azabicyclo[3.2.1] octan-3-ol] is the prototypical monocyclic tropane nucleus bearing a single hydroxyl at C-3 [87]. It exists as a white, hygroscopic crystalline solid (mp 64 °C; bp 233 °C) that is highly soluble in water and polar organic solvents [87,115]. In *Datura* spp. (e.g., *D. stramonium*, *D. metel*), tropine arises biosynthetically from tropinone via stereospecific reduction by tropinone reductase I (TRI), following a series of upstream transformations—putrescine, N-methylputrescine, N-methyl-Δ^1^-pyrrolinium, hygrine, tropinone—catalyzed by ornithine decarboxylase, putrescine N-methyltransferase, N-methylputrescine oxidase, and nonenzymatic rearrangements. Tropine accumulates primarily in roots (up to 0.5 mg·g^−1^ dry weight) and to a lesser extent in stems and leaves (0.1–0.2 mg·g^−1^) of mature *Datura* plants, reflecting its role as a key intermediate for downstream esters (e.g., littorine, hyoscyamine). [43,115,116]

Toxinology of Tropine

As a simple alcohol rather than an ester, tropine exhibits negligible affinity for muscarinic acetylcholine receptors and does not elicit classical anticholinergic signs. Acute toxicity studies report an oral LD_50_ in rats > 2000 mg·kg^−1^, a mouse intraperitoneal LD_50_ ~ 139 mg·kg^−1^, and an intravenous LD_50_ in rabbits > 50 mg·kg^−1^, indicating a wide safety margin relative to tropane esters like atropine (LD_50_ ~ 6 mg·kg^−1^ in humans). Rapid renal clearance (t ½ ≈ 1–2 h), efficient hepatic metabolism to tropanol metabolites, and poor blood–brain barrier penetration underlie tropine’s low systemic and central toxicity. No human poisonings have been attributed to isolated tropine [31,33,36,117].

Pharmaceutical and Synthetic Applications of Tropine

Although tropine itself lacks direct therapeutic indications, it is indispensable as a chemical building block in the industrial and laboratory synthesis of clinically meaningful tropane esters. Tropine’s primary pharmaceutical utility lies in its N-alkylation and esterification reactions:The classic synthesis of atropine begins with tropine, which is esterified at C-3 with tropic acid (3-hydroxy-2-phenylpropanoic acid). In practice, tropic acid is first converted to its acid chloride (SOCl_2_, catalytic DMF, 0 °C), then coupled to tropine in dry dichloromethane with pyridine as the base to give racemic atropine in 70–85% yield. Resolution of the racemate into (–)-hyoscyamine and its enantiomer is accomplished by formation of diastereomeric salts—typically with (–)-tartaric acid in ethanol—followed by fractional crystallization. Filtration and basification liberate enantiopure (S)-hyoscyamine, which exhibits ≈approximately 50–100 times greater affinity for muscarinic receptors than its (R) counterpart [41,117,118].Scopolamine can be generated from hyoscyamine via selective epoxidation of the C-6, C-7 bond. Chemically, this is achieved by treating the hyoscyamine free base with a peracid (e.g., m-CPBA) in chloroform at 0 °C, yielding the 6β,7β-epoxide in 50–70% yield after silica gel chromatography. Alternatively, biosynthetic conversion employs hyoscyamine 6β-hydroxylase (H6H) in recombinant microbial or plant-cell systems, where NADPH and O_2_ drive sequential hydroxylation and intramolecular epoxide closure, routinely achieving over 90% conversion under optimized fermentation conditions. Scopolamine’s rigid epoxide ring enhances central nervous system penetration and receptor affinity, underpinning its superior antiemetic potency [38,119].Quaternization of tropine’s bridgehead nitrogen provides a route to inhaled anticholinergics with minimal systemic exposure. For ipratropium bromide, tropine is stirred with excess isopropyl bromide in dry acetone at reflux for 12–24 h, yielding the isopropyl quaternary ammonium salt in an isolated yield of 60–75%. Recrystallization from ethanol affords the pharmaceutically standardized monohydrate. Tiotropium bromide is prepared analogously by SN2 alkylation with 2-thienylmethyl chloride, followed by the introduction of two thienyl rings via subsequent alkylation steps, and finally, counterion exchange to the bromide salt—overall yields for the multi-step sequence range from 45% to 55%. The permanent positive charge of these agents prevents blood–brain barrier crossing, thereby focusing their M_3_-selective antagonism on bronchial smooth muscle for the treatment of COPD and asthma [120,121].Tropine metabolites serve as lead scaffolds in medicinal chemistry campaigns targeting central and peripheral nervous system disorders. By modifying the C-3 hydroxyl group (e.g., carbamate linkages, ether conjugates), researchers have generated novel compounds with tailored muscarinic subtype selectivity and pharmacokinetic profiles, underscoring tropine’s versatility as a synthetic and pharmacophoric template [43,122,123].

Table 2 summarizes the minor and less common *Datura* alkaloids, hygrine, apoatropine, anisodine, and tropine, explaining the underlying scaffold common to tropane alkaloids., the plant tissues where each alkaloid is most abundantly found, the key chemical change(s) relative to the unmodified tropane skeleton, and the main therapeutic application (for pharmacologically active compounds) or the principal analytical/research utility (for intermediates or markers).

### 2.3. Synthetic and Semi-Synthetic Derivatives

Anticholinergic drugs such as atropine and scopolamine are designed based on the tropane series derived from *Datura* alkaloids. These compounds have been modified to reduce central nervous system penetration, increase drug selectivity, or prolong their duration of action. These drugs, which are classified as quaternary ammonium derivatives, have limited central nervous system penetration due to their permanent positive charge. Tertiary amine derivatives, on the other hand, have limited central nervous system penetration; triamine derivatives, however, cross the blood–brain barrier in central states as they are lipophilic [38,55,124].

#### 2.3.1. Quaternary Ammonium Derivatives

The four quaternary ammonium derivatives, Ipratropium Bromide, Tiotropium Bromide, Methscopolamine, and Trospium Chloride (compounds **12**–**15**, respectively, in Figure 3) are all structurally related to scopolamine and share a tropane-based backbone, yet differ in their functional modifications and stereochemistry [125,126,127,128]. Each compound incorporates a quaternary ammonium group that enhances water solubility and restricts penetration into the central nervous system. Stereochemically, they feature one or more chiral centers, with ipratropium and tiotropium exhibiting defined enantiomeric preferences, the (R, R)-isomer of ipratropium being pharmacologically dominant. A thiophene moiety further distinguishes tiotropium, while trospium contains a benzilic acid ester group that favors M3 receptor targeting. Methscopolamine retains the characteristic epoxide bridge seen in scopolamine but adds a methyl group on the nitrogen to form the quaternary salt [55,81,129].

All four compounds are synthesized via the derivatization of scopolamine or its tropine scaffold. Ipratropium is produced by quaternization with a bromide ion, whereas tiotropium is developed through structural refinement of ipratropium, notably introducing a thiophene ring, which prolongs receptor binding [130,131]. Methscopolamine results from direct N-methylation of scopolamine, forming a stable quaternary ammonium ion [132]. Trospium chloride, synthesized from the condensation of tropine and benzilic acid followed by quaternization, adopts a configuration that limits CNS access and favors peripheral action [133]. These structural distinctions significantly influence their receptor selectivity, pharmacokinetics, and safety. Notably, all four compounds achieve improved peripheral specificity and reduced CNS side effects compared to scopolamine due to their quaternary structures [126,127,128].

The toxicological profiles of these compounds reflect their chemical design. Ipratropium and tiotropium, when used as inhaled agents, exhibit minimal systemic toxicity and limited CNS effects [134,135,136]. Methscopolamine has a relatively high LD_50_ and low CNS penetration, with toxicity manifesting primarily as peripheral anticholinergic symptoms such as xerostomia and blurred vision [137]. Trospium chloride shares this favorable safety profile, with most adverse effects being mild and related to its antimuscarinic action [43]. Importantly, the quaternary ammonium nature of all four compounds plays a pivotal role in reducing the risks associated with central anticholinergic toxicity, a common concern with scopolamine [41,43,46].

Pharmacologically, these agents are utilized for distinct but overlapping indications based on their tissue-specific actions. Ipratropium and tiotropium are used in respiratory medicine as bronchodilators for conditions such as chronic obstructive pulmonary disease (COPD) and asthma, with tiotropium offering a longer duration of action, making it suitable for once-daily dosing [138,139]. Methscopolamine is indicated for gastrointestinal disorders, including peptic ulcers and excessive salivation, leveraging its peripheral antimuscarinic effects while minimizing central effects [127,132]. Trospium chloride is approved for urological use in managing overactive bladder, demonstrating strong efficacy in reducing urinary urgency and incontinence [140,141]. The clinical success of these agents exemplifies the utility of quaternary ammonium derivatization in enhancing therapeutic index and safety by optimizing peripheral receptor selectivity.

#### 2.3.2. Tertiary (Lipophilic) Amine Derivatives

The tertiary (lipophilic) amine derivatives of atropine, Homatropine hydrobromide, Tropicamide, Cyclopentolate, Oxybutynin, Tolterodine, and Trihexyphenidyl (compounds **16**–**21**, respectively, in Figure 4), share essential structural motifs derived from tropane pharmacophores but exhibit diverse chemical architectures and stereochemical profiles. Homatropine retains the tropane nucleus, esterified with mandelic acid, maintaining one chiral center within a rigid bicyclic system [142,143]. Tropicamide and Cyclopentolate preserve the tropic acid-derived chiral α-hydroxy center but replace the tropane base with linear or cyclic amines, respectively, introducing increased conformational flexibility [143]. Oxybutynin, though derived from tropic acid analogs, includes a cyclohexyl group and a diethylamino butynyl side chain [144], while Tolterodine’s phenylpropanolamine scaffold features a central stereocenter critical for its enantioselective receptor binding [145]. Trihexyphenidyl, with a tertiary alcohol bearing cyclohexyl and phenyl rings, is highly lipophilic and CNS-penetrant; its single stereocenter is of limited clinical consequence due to racemic formulation. These compounds exhibit substantial variability in steric bulk, flexibility, and amine basicity, all of which contribute to their pharmacodynamic profiles [146].

The synthesis of these compounds often begins with tropine or phenylacetic acid derivatives. Homatropine is obtained by esterification of tropine with mandelic acid, producing a less potent but more rapidly cleared analog of atropine [142]. Tropicamide replaces atropine’s ester bond with a more hydrolytically stable amide by condensing tropic acid with a pyridyl-ethylamine, increasing clearance and limiting systemic effects [143]. Cyclopentolate follows a similar esterification route using a cyclopentyl-substituted tropate acid and dimethylaminoethanol [143]. Oxybutynin and Tolterodine, structurally divergent, maintain functional similarity via a tertiary amine and aromatic rings; their synthesis involves alkylation of hydroxyaryl acetic acid cores with amino alcohols or amines [144,145]. Trihexyphenidyl is synthesized through the condensation of a benzhydryl alcohol with piperidine, retaining receptor affinity through both hydrophobic and ionic interactions [146]. Across all compounds, SAR analysis emphasizes the retention of a basic amine, flexible or lipophilic hydrophobic groups, and a hydrogen bond acceptor/donor near the receptor interaction site, all of which are crucial for selective muscarinic antagonism. The structural modifications generally reduce duration of action, enhance CNS selectivity, or increase peripheral receptor specificity [144,145,146].

From a toxicological perspective, these tertiary amine derivatives show varying CNS and systemic toxicity, primarily influenced by their ability to cross the blood–brain barrier and their muscarinic receptor selectivity. Homatropine, though CNS-penetrant, has reduced potency and shorter duration than atropine, resulting in milder toxicity [147,148]. Tropicamide and Cyclopentolate can induce transient CNS effects such as agitation or hallucinations in pediatric or sensitive populations, though systemic toxicity is rare when used ophthalmically [149,150]. Oxybutynin, especially in oral form, may produce cognitive dysfunction, hallucinations, or confusion, particularly in elderly patients, though transdermal administration reduces peak plasma concentrations and mitigates risk [151,152,153]. Tolterodine has a more favorable CNS profile due to reduced brain penetration and active metabolite (5-HMT) formation, while Trihexyphenidyl, though effective centrally, carries a higher burden of cognitive and behavioral toxicity, including potential for abuse and psychosis in high doses. Most agents have LD_50_ values in the moderate range (200–1000 mg/kg, oral, rat), underscoring the need for dose titration and monitoring in vulnerable populations [145,146].

Pharmacologically, these compounds are employed in diverse therapeutic contexts. Homatropine is primarily used as a short-acting mydriatic and adjunct in cough suppressants [149]. Tropicamide and Cyclopentolate are both staples in ophthalmology for diagnostic mydriasis and cycloplegia, with Cyclopentolate preferred in pediatric settings due to its intermediate duration [154,155]. Oxybutynin is widely used in the management of overactive bladder and neurogenic detrusor overactivity, and is available in oral, transdermal, and topical forms [156,157]. Tolterodine, which offers improved tolerability and reduced cognitive side effects, is often preferred in elderly patients for managing urinary frequency and urgency [158]. Trihexyphenidyl is used in neurology for managing Parkinsonian tremor and extrapyramidal symptoms, though its CNS side effect profile limits long-term use in older patients. Together, these tertiary amine derivatives provide a broad toolkit for modulating cholinergic tone across organ systems, optimized through chemical tailoring of tropane-inspired scaffolds [149]. Table 3 shows comparisons for both tertiary (lipophilic) amine and quaternary ammonium antimuscarinic agents. It includes structural features, CNS penetration, clinical uses, and notable toxicity profiles.

### 2.4. Other Alkaloid Classes and Their Datura Metabolites

While tropane alkaloids dominate *Datura’s* chemistry, other alkaloid classes have been reported in trace amounts:

#### 2.4.1. Harmane and Norharmane

Harmane and Norharmane (1-methyl-9H-pyrido[3,4-b]indole and 9H-pyrido[3,4-b]indole, compounds **22** and **23** in Figure 4) are structurally related β-carboline alkaloids found in *DD. stramonium* and various cooked foods [159,160,161,162]. Harmane is highly lipophilic and accumulates in brain tissue at concentrations up to 55-fold higher than plasma levels [159]. It exhibits low oral bioavailability (~19%) and undergoes rapid metabolism via sulphation and glucuronidation [163]. Norharmane shows similar distribution patterns and is endogenously present in human plasma and platelets [162]. At high doses, both compounds may induce tremor, impair working memory, and exhibit neurotoxicity, necessitating careful dose optimization and formulation strategies [159,162].

Both compounds are potent inhibitors of monoamine oxidase A (MAO-A) and MAO-B, with IC_50_ values ranging from 0.5 to 6.5 μM, depending on the isoform and compound [159,160,161,162]. Harmane enhances dopaminergic and serotonergic transmission, potentiating nicotine reinforcement and modulating mood-related behaviors [159]. Norharmane similarly induces antidepressant-like effects in murine models, reducing locomotor activity and anxiety at doses of 2.5–10 mg/kg intraperitoneally [162]. Both alkaloids interact with imidazoline, benzodiazepine, and opioid receptors, contributing to their anxiolytic, analgesic, and anticonvulsant properties. Harmane has also been implicated in the pathogenesis of essential tremor, with elevated brain concentrations linked to dopaminergic disruption [159,162].

In addition, Harmane and Norharmane exhibit dose-dependent cytotoxicity against various human cancer cell lines. Harmane induces apoptosis in MCF-7 cells via mitochondrial depolarization, PARP1 inhibition, and Bax activation, independent of p53 signaling [164]. Norharmane acts as a photosensitizer, enhancing DNA damage and cytotoxicity under UV exposure, particularly in HeLa and SH-SY5Y cells [160]. In *Caenorhabditis elegans*, harmane increases survival during bacterial infection by stimulating innate immunity, while Norharmane modulates quinolinic acid and kynurenine pathways in THP-1 monocytes, suggesting IDO inhibition and immunoregulatory potential [162].

In addition, harmane disrupts bacterial biofilms, damages peptidoglycan structures, and impairs ribosomal function, showing efficacy against *E. coli* O157:H7 and *Staphylococcus aureus* [164]. Norharmane enhances the activity of antibiotics such as polymyxin B, levofloxacin, and imipenem against drug-resistant *Pseudomonas aeruginosa*, acting as an antibiotic adjuvant by increasing membrane permeability and suppressing quorum sensing [165].

Both compounds inhibit virulence factor production and biofilm formation in *Serratia marcescens*, with Norharmane showing superior inhibition (53.9%) compared to Harmane (19.9%) [159,162].

On the other hand, both exhibit antioxidant activity by scavenging reactive oxygen species and enhancing the activity of endogenous antioxidant enzymes. Harmane protects neuronal tissues from oxidative stress and supports mitochondrial integrity [166]. Norharmane modulates polar auxin transport in Arabidopsis thaliana by inhibiting PIN2, PIN3, and PIN7, suggesting broader implications in redox signaling and plant pharmacology [167].

#### 2.4.2. Daturafolisides

Daturafolisides A–I (compounds **24–32** in Figure 5) are a group of structurally novel withanolide glycosides isolated from the leaves of *D. metel* L. These compounds belong to the C28 steroidal lactone family and are characterized by an ergostane-based skeleton with various oxygenation patterns and glycosidic linkages [168]. Their pharmacological relevance stems from their anti-inflammatory, cytotoxic, and immunomodulatory activities [169]. As anti-inflammatory activity, all nine Daturafolisides were evaluated for their ability to inhibit nitric oxide (NO) production in LPS-stimulated RAW 264.7 murine macrophages, a standard in vitro model for inflammation. Among them, Daturafolisides A and B exhibited the most potent inhibitory effects, with IC_50_ values of 20.9 μM and 17.7 μM, respectively [168,169]. These compounds significantly suppressed iNOS expression and reduced nitrite accumulation, indicating their potential to modulate macrophage-mediated inflammatory responses. Moderate activity was observed for Daturafolisides C, D, F, and G (IC_50_ ranging from 52.8 to 71.2 μM), while the remaining analogs showed weaker effects [168,169].

Besides the anti-inflammatory activity, daturafolisides demonstrated cytotoxicity against human cancer cell lines, including A549 (lung), BGC-823 (gastric), and K562 (leukemia). In particular, daturafoliside B exhibited IC_50_ values ranging from 0.05 to 3.5 μM, indicating high potency [169]. Mechanistic studies revealed that these compounds induce apoptosis via mitochondrial pathways, disrupt cell cycle progression, and inhibit DNA synthesis. Their selective cytotoxicity and low toxicity to non-transformed cells support further investigation as anticancer agents [169].

Furthermore, daturafolisides have also been implicated in immune regulation, particularly in models of psoriatic skin inflammation [170]. In vivo studies using imiquimod-induced dermatitis in mice showed that total withanolide fractions containing daturafolisides reduced epidermal hyperplasia, normalized spleen and thymus metabolite profiles, and downregulated pro-inflammatory cytokines such as IL-17 and TNF-α. These findings suggest potential applications in autoimmune skin disorders and topical immunosuppressive therapies [170,171].

#### 2.4.3. Daturataturin

Daturataturin A and B (compounds **33** and **34** in Figure 5) are structurally distinct withanolide glycosides isolated from the flowers and pericarps of *D. metel* [172]. These compounds belong to the ergostane-type steroidal lactone family and are characterized by hydroxylated and glycosylated moieties at key positions on the withanolide backbone [172,173]. Metabolite identification studies in rats revealed that daturataturin A undergoes hydroxylation, methylation, and glucuronidation, producing multiple phase I and phase II metabolites detectable in plasma, urine, and feces [174,175]. The presence of methylated and sulfated derivatives suggests active hepatic metabolism and potential for oral bioavailability optimization. These pharmacokinetic insights are critical for future formulation and dosing strategies [174,175].

Daturataturin A has demonstrated potent anti-inflammatory activity in human keratinocyte (HaCaT) models. It induces autophagy via the PI3K-Akt-mTOR signaling pathway, leading to cell cycle arrest and senescence in hyperproliferative epidermal cells [175]. This autophagy-mediated suppression of inflammation suggests a mechanistic basis for its traditional use in treating psoriasis and other inflammatory skin conditions. Daturataturin A also downregulates pro-inflammatory cytokines and inhibits NF-κB activation, further supporting its role in immune modulation [172].

Moreover, both daturataturin A and B have been evaluated for their immunosuppressive effects on RAW264.7 macrophages [172,176]. Daturataturin B, in particular, exhibited significant inhibition of nitric oxide production with an IC_50_ of 38.3 μM, indicating suppression of macrophage activation [172]. Western blot analysis revealed downregulation of p-ERK, p-JNK, p-p38, and NF-κB p65, confirming their impact on key inflammatory signaling pathways [172]. These findings suggest potential applications in autoimmune diseases and chronic inflammatory disorders [172,176].

In addition, in vitro studies have shown that daturataturin A exhibits cytotoxic activity against human cancer cell lines, such as MDA-MB-435 and SW-620, with IC_50_ values in the low micromolar range [172,177]. It induces apoptosis and inhibits proliferation by modulating mitochondrial pathways and reducing oxidative stress. These effects position daturataturin A as a candidate for oncologic drug development, particularly in hormone-sensitive and epithelial-derived tumors [177].

#### 2.4.4. Withametelins

Withametelins I–P are a series of structurally diverse withanolide-type steroidal lactones isolated from the methanolic extract of *D. metel* flowers. These compounds exhibit a unique C21–O–C24 ether bridge and a bicyclic lactone side chain, often featuring an exocyclic double bond at C25–C27 [176]. ADME profiling indicates favorable oral bioavailability, GI absorption, and plasma protein binding. Withametelins are metabolized by CYP1A2, CYP2C19, and CYP3A4, and show non-carcinogenicity in rodent models. Acute and subacute toxicity studies in rats revealed a NOAEL of 1.25 mg/kg/day, with higher doses causing thyroid dysregulation and hepatic enzyme elevation [178,179,180].

Several Withametelins (notably I, J, and M) have demonstrated potent cytotoxicity against human cancer cell lines, including DU145 (prostate) and HepG2 (liver) [181,182]. IC_50_ values range from 7.6 to 15 μM, indicating strong antiproliferative effects. Mechanistically, these compounds induce apoptosis via mitochondrial disruption, ROS generation, and NF-κB inhibition, suggesting their utility in chemotherapeutic regimens [183,184,185].

In addition, withametelins P and N have shown significant inhibition of nitric oxide (NO) production in LPS-activated RAW264.7 macrophages, with IC_50_ values below 20 μM [186,187]. This suppression is linked to the downregulation of COX-2, iNOS, and MAPK signaling pathways, indicating their potential in treating conditions such as psoriasis, rheumatoid arthritis, and inflammatory bowel disease [186,187,188].

Furthermore, in vitro DPPH assays revealed that Withametelins possess free radical scavenging activity, with IC_50_ values comparable to those of standard antioxidants, such as ascorbic acid. Their ability to cross the blood–brain barrier and modulate the Nrf2/Keap1/HO-1 pathways suggests neuroprotective roles in Alzheimer’s disease, Parkinson’s disease, and neuropathic pain [189,190].

Besides that, withametelins I and K exhibit broad-spectrum protein kinase inhibition, particularly targeting NF-κB, MAPK, and cyclin-dependent kinases (CDKs). These interactions were confirmed through molecular docking studies, with binding energies ranging from –11.3 to –7.8 kcal/mol, supporting their role in modulating signal transduction and cell cycle arrest [191,192].

#### 2.4.5. Daturametelin J

Daturametelin J (compound **35** in Figure 5) is a withanolide-type steroidal lactone glycoside isolated from the aerial parts of *D. metel L*. Structurally, it features a 1-oxowitha-2,5,24-trienolide backbone with hydroxyl substitutions and glycosidic linkages that contribute to its pharmacological versatility [171]. The metabolite profiling in rats revealed that daturametelin J undergoes hydroxylation, methylation, and glucuronidation, producing phase I and II metabolites detectable in plasma, urine, and feces. The methylated derivative daturametelin L was identified as a major metabolite, indicating hepatic biotransformation. These insights are crucial for optimizing oral bioavailability, formulation strategies, and dose–response modeling [171,174].

As part of the broader daturametelin family, compound J has attracted attention for its bioactivity. It has demonstrated inhibition of nitric oxide (NO) production in LPS-stimulated RAW264.7 macrophages, suggesting suppression of innate immune activation [169,193]. It downregulates COX-2, iNOS, and NF-κB p65, indicating its potential to modulate pro-inflammatory signaling cascades. These effects support its traditional use in treating psoriasis, eczema, and rheumatoid arthritis, where immune dysregulation plays a central role [169,170,193].

Additionally, in vitro assays have shown that daturametelin J exhibits cytotoxicity against human cancer cell lines, including A549 (lung) and K562 (leukemia), with IC_50_ values in the low micromolar range [194,195]. Mechanistic studies suggest that it induces apoptosis via mitochondrial disruption, ROS generation, and cell cycle arrest at G2/M phase. These findings position daturametelin J as a promising scaffold for the development of anticancer drugs, particularly in hematologic and epithelial malignancies [194,195].

In addition, daturametelin J has demonstrated free radical scavenging activity in DPPH and ABTS assays, with IC_50_ values comparable to those of standard antioxidants. Its ability to activate the Nrf2/HO-1 pathway and reduce oxidative stress in neuronal models suggests potential applications in Alzheimer’s disease, Parkinson’s disease, and ischemic stroke. These neuroprotective effects are attributed to its phenolic and lactonic moieties [12,196]. Table 4 provides a comparative overview of the principal pharmacological characteristics and therapeutic uses of the compounds discussed.

## 3. Future Directions and Challenges

The therapeutic potential of *Datura*-derived alkaloids is vast; however, significant challenges remain in fully unlocking their biomedical applications. While advances in bioorganic chemistry have elucidated the structural features and pharmacological profiles of these compounds, further research is necessary to bridge the existing gaps in understanding their complex biosynthesis, toxicology, and clinical efficacy [53,197]. In the following subsections, we outline several key areas for future exploration:

### 3.1. Metabolic Pathway Elucidation

Despite considerable progress, the complete biosynthetic pathways of *Datura* alkaloids, especially minor derivatives such as hygrine and cuscohygrine, remain inadequately characterized [53]. The discovery of enzymes involved in the conversion of precursor molecules into bioactive alkaloids opens exciting possibilities for the metabolic engineering of these compounds. Future research should focus on employing advanced biochemical tools, such as CRISPR/Cas9 gene editing and synthetic biology approaches, to dissect the enzymatic cascades responsible for tropane alkaloid synthesis [198,199]. Additionally, understanding the regulatory mechanisms governing these pathways will be crucial for optimizing the production of desired alkaloids, potentially overcoming the issue of low yield in natural sources [198,199].

### 3.2. Pharmacological Profiling and Toxicology

The bioactivity of *Datura* alkaloids is heavily influenced by their interaction with muscarinic receptors and other molecular targets. However, the pharmacological profiles of many minor alkaloids, such as hygrine and apoatropine, have been insufficiently explored [55,200]. It is crucial to expand research on these compounds to identify potential therapeutic applications in neuropharmacology, infectious diseases, and psychiatric disorders. Moreover, the narrow therapeutic index and toxicological risks associated with these alkaloids pose a challenge. Future studies should aim to develop safer derivatives with enhanced receptor selectivity, minimal side effects, and improved bioavailability [196,200]. Animal models and clinical trials will be essential for assessing the toxicity thresholds of these compounds and their long-term safety profiles [55,196,200].

### 3.3. Pharmacogenomics and Personalized Medicine

A promising avenue of future research lies in the application of pharmacogenomics to understand how individual genetic variations influence the metabolism and effectiveness of *Datura*-derived alkaloids. For instance, variations in cytochrome P450 enzymes may lead to differential responses to scopolamine or atropine. Personalized medicine approaches could optimize dosing strategies based on genetic profiling, reducing adverse effects while maximizing therapeutic benefits. This direction is particularly relevant in light of the growing interest in targeted drug therapies for neurological disorders [199,201,202].

### 3.4. Synthetic Derivatives and Analog Development

The semi-synthetic modifications of *Datura* alkaloids have already proven fruitful in the development of drugs like ipratropium and tiotropium for treating respiratory conditions. However, there is room for further innovation in designing new analogs with enhanced specificity and fewer side effects. Future drug development may focus on refining the pharmacokinetic properties of these compounds through modifications to their chemical structures, aiming to improve their selectivity for specific receptor subtypes or tissue targeting [53,87,200].

### 3.5. Ethnobotanical and Traditional Use Validation

*Datura* species have a long history of use in traditional medicine across various cultures. However, many of these uses lack robust scientific validation. Systematic clinical studies are needed to confirm the efficacy of *Datura* alkaloids in treating conditions such as asthma, motion sickness, and gastrointestinal disorders. In parallel, the ethnobotanical knowledge of indigenous communities may provide valuable insights into new applications for *Datura* metabolites, particularly those that have not been extensively studied for their potential as pharmaceuticals [55,203,204].

### 3.6. Regulatory and Ethical Considerations

The inherent toxicity of *Datura* alkaloids, particularly at higher doses, raises significant regulatory and ethical concerns in their development as pharmaceuticals. The controlled nature of many *Datura* compounds, alongside the associated risks of misuse and overdose, necessitates rigorous regulation and oversight. Future efforts must focus on establishing clear regulatory frameworks for the safe incorporation of *Datura* metabolites into clinical practice. Furthermore, ethical considerations regarding the sourcing, extraction, and potential misuse of these potent compounds must be prioritized [55,196,205].

## 4. Conclusions

Research on *Datura* alkaloids has increased dramatically during the last ten years, exposing the genus’s pharmacological range and chemical variety. The distribution of tropane and non-tropane alkaloids across species, plant organs, and developmental phases has been clarified by advances in metabolomic profiling and high-resolution analytical techniques. While pharmacological research has validated proven uses of key tropanes and highlighted the bioactivities of lesser-known metabolites, biosynthetic studies have expanded the understanding of enzyme routes and genetic regulation. These discoveries have improved the foundation for safe and effective medicinal or agricultural applications and reinforced the chemotaxonomic framework for *Datura*. Going forward, there will be more chances to enhance molecules isolated from *Datura* for therapeutic, biochemical, and forensic applications thanks to advancements in synthetic biology, structure-guided medication design, and advanced toxicological evaluation. Standardizing analytical techniques, identifying genus-specific chemical markers, and developing safer derivatives or formulations with lower toxicity and a better therapeutic index are examples of translational priorities. Assessing the broader effects of *Datura* metabolites in both natural and applied contexts will require combining ecological and agronomic data, such as allelopathic and bioherbicidal capabilities.

## Figures and Tables

**Figure 1 toxins-17-00469-f001:**
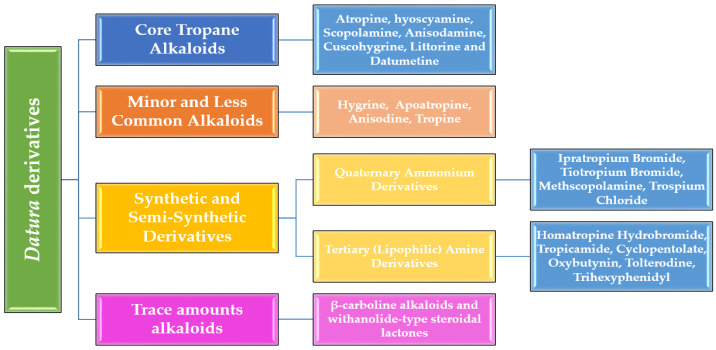
Classification of alkaloids from *Datura* species.

**Figure 2 toxins-17-00469-f002:**
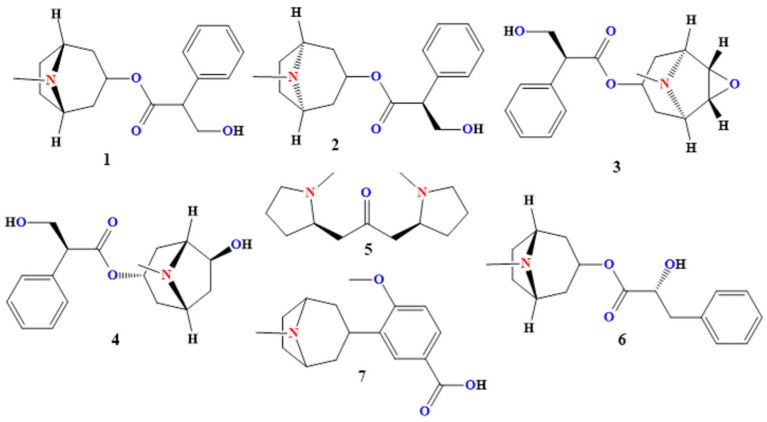
Chemical structures of (**1**) atropine, (**2**) hyoscyamine, (**3**) scopolamine, (**4**) anisodamine, (**5**) cuscohygrine, (**6**) littorine, and (**7**) datumetine.

**Figure 3 toxins-17-00469-f003:**
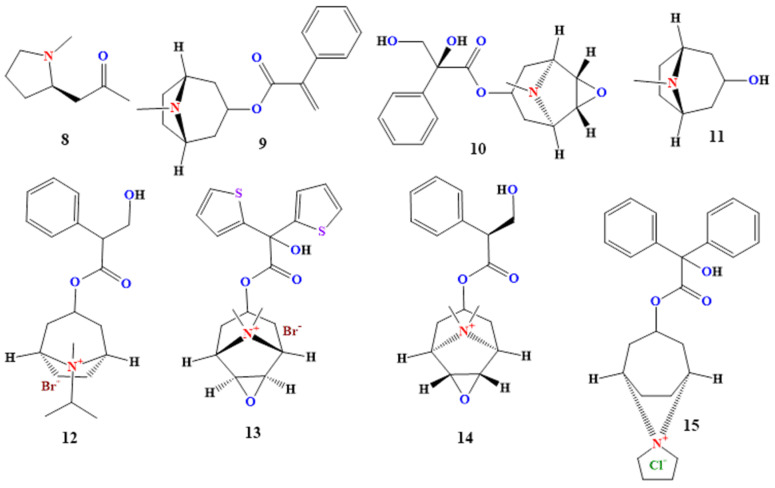
Chemical structures of (**8**) hygrine, (**9**) apoatropine, (**10**) anisodine, (**11**) tropine, (**12**) ipratropium bromide, (**13**) tiotropium bromide, (**14**) methscopolamine, and (**15**) trospium chloride.

**Figure 4 toxins-17-00469-f004:**
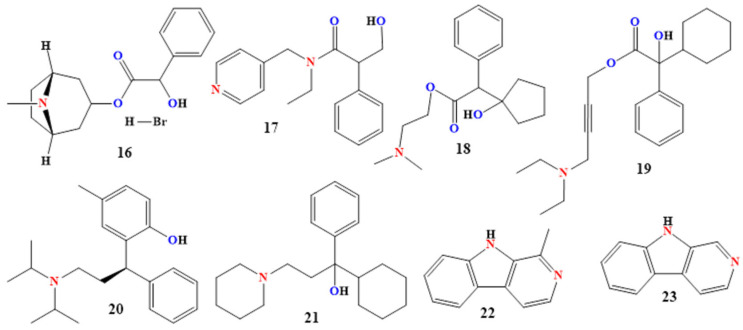
Chemical structures of (**16**) homatropine hydrobromide, (**17**) tropicamide, (**18**) cyclopentolate, (**19**) oxybutynin, (**20**) tolterodine, (**21**) trihexyphenidyl, (**22**) harmane, (**23**) norharmane.

**Figure 5 toxins-17-00469-f005:**
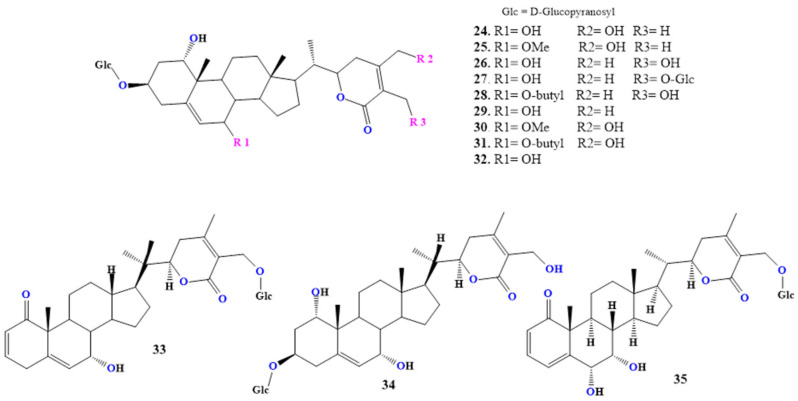
Chemical structures of (**24–32**) daturafolisides A–I, (**33**) daturataturin A, (**34**) daturataturin B, (**35**) daturametelin J.

**Table 1 toxins-17-00469-t001:** Summary of the principal *Datura*-derived tropane alkaloids, highlighting each compound’s core scaffold, plant-part occurrence, key structural modification(s), and main application or utility.

Compound	Core Structure	Occurrence	Structural Modification	Primary Use/Utility	Distribution
Atropine	Tropane (7-azabicyclo[3.2.1]octane) esterified with tropic acid	Major alkaloid in seeds and leaves of *D. stramonium*, *D. metel*	Racemic mixture of (S)- and (R)-hyoscyamine enantiomers	Organophosphate antidote; preanesthetic antisialagogue; mydriatic/cycloplegic	Widespread—reported in multiple Solanaceae genera (*Atropa, Hyoscyamus, Scopolia, Duboisia, Datura*)
Hyoscyamine	Tropane core + tropic acid ester	Predominates in young leaves, seeds and roots of *D. stramonium*/*D. metel*	Pure (S)-enantiomer of hyoscyamine (levo-form)	GI antispasmodic (IBS, colic); adjunct in Parkinson’s; visceral pain relief	Widespread—multiple Solanaceae (*Atropa, Hyoscyamus, Scopolia, Duboisia, Datura*)
Scopolamine	Tropane core + tropic acid ester + 6β-epoxide bridge	High in flowers and young leaves of *D. stramonium*, *D. innoxia*	6β-Epoxide on the hyoscyamine backbone	Motion sickness prophylaxis (TTS patch); PONV antiemetic; preanesthetic; ophthalmic mydriatic	Widespread—multiple *Solanaceae genera*
Anisodamine	Tropane core + tropic acid ester + 6β-hydroxyl	Trace in stems, leaves and seeds of *D. stramonium*	Addition of C-6β hydroxyl to hyoscyamine	Septic shock therapy (China); slowing of myopia progression ophthalmically	Widespread—reported in *Anisodus, Hyoscyamus, Datura*
Cuscohygrine	Bis-N-methylpyrrolidine rings linked by a propanone bridge	Trace alkaloid in seeds, roots and leaves of *Datura* spp.	Lacks bicyclic tropane core; two N-methylpyrrolidine moieties	Analytical/chemotaxonomic marker in alkaloid profiling; biosynthetic intermediate	Widespread—Solanaceae and Erythroxylaceae (*Erythroxylum coca*)
Littorine	Tropane core + phenyllactic acid ester	Concentrated in the roots of *D. stramonium*; lower in the aerial parts	Tropine esterified with phenyllactic (instead of tropic) acid	Biosynthetic precursor to hyoscyamine; metabolic tracer; species authentication	Widespread—multiple tropane-producing Solanaceae
Datumetine	Tropane core + 4-methoxybenzoic acid ester	Trace in leaves of *D. metel*	Ester linkage to 4-methoxybenzoate rather than tropic acid	Research tool as NMDAR modulator; neuroprotective/excitotoxicity studies	Reported primarily from *Datura* (esp. D. metel); no records in other genera to date

**Table 2 toxins-17-00469-t002:** Summary of the minor and less common *Datura* alkaloids, highlighting their core scaffold, plant occurrence, defining structural feature(s), and principal research or analytical utility.

Compound	Core Structure	Occurrence	Structural Modification	Primary Use/Utility
Hygrine	N-Methylpyrrolidin-2-one (monopyrrolidine ring + ketone side chain)	Trace in seeds, roots and young leaves (0.02–0.1 mg/g DW *)	Lacks bicyclic tropane; simple pyrrolidine with an acetone moiety	Early biosynthetic intermediate; stable isotope tracer; analytical internal standard for HPLC–MS profiling
Apoatropine	Tropine core esterified to 2-phenylprop-2-enoic (atropic) acid	Minor in stems and leaves of *Datura* spp.	An unsaturated atropic acid ester replaces tropic acid	Structure–activity probe in tropane SAR studies; analytical marker differentiating tropane esters
Anisodine	7-Azatricyclo[3.2.1.0^2,4^]-nonane core with 3-oxa epoxide + benzoate ester	Trace in flowers, leaves and seeds (≤ 0.05 mg/g DW *)	Epoxide bridge (C-6→C-7) and esterified to α-hydroxybenzoic acid	Research tool for M_1_–M_5_/α_1_ receptor pharmacology; template for anticholinergic–vasodilator hybrids
Tropine	7-Azabicyclo[3.2.1]octan-3-ol (tropane ring + C-3 hydroxyl)	Abundant in roots (0.3–0.5 mg/g DW *) & aerial parts (0.1–0.2 mg/g DW *)	De-esterified tropane (no acyl side chain)	Universal precursor for tropane esters (atropine, scopolamine); synthetic scaffold for inhaled/quaternary anticholinergics; analytical standard

* DW = dry weight.

**Table 3 toxins-17-00469-t003:** Comparative Structural and Pharmacological Profile of Tertiary and Quaternary Antimuscarinic Agents.

Group	Compound	Key Structural Features	CNS Penetration	Main Use	Mechanism of Action
Quaternary Ammonium Derivatives	Ipratropium Bromide	Tropane derivative; isopropyl on ester	Minimal	COPD	Non-selective muscarinic antagonist; peripheral bronchodilation
	Tiotropium Bromide	thiophene rings; tropane core	Minimal	Long-acting COPD	Selective M3 antagonist; prolonged bronchodilation
	Methscopolamine	6,7-epoxide tropane; scopolamine-derived	Minimal	Peptic ulcers; GI spasms	Peripheral muscarinic antagonist; antisecretory and antispasmodic
	Trospium Chloride	benzilic acid ester with a tropane backbone	Minimal	Overactive bladder; urinary frequency	M3-preferring antagonist; inhibits detrusor overactivity
Tertiary Amine Derivatives	Homatropine hydrobromide	Tropine and mandelic acid ester	Moderate	Ophthalmic mydriasis; cough suppressant adjunct	Non-selective muscarinic antagonist; central and peripheral action
	Tropicamide	Tropic acid and pyridyl amide	Low to moderate	Diagnostic mydriasis/cycloplegia	Non-selective muscarinic antagonist; rapid receptor dissociation
	Cyclopentolate	Cyclopentyl tropate ester and dimethylaminoethanol	Moderate	Pediatric eye exams; diagnostic cycloplegia	Muscarinic antagonist; short-acting cycloplegic
	Oxybutynin (Oxytrol)	Cyclohexyl phenyl ester and butynyl diethylamino chain	High (oral); Low (patch)	Overactive bladder; neurogenic bladder	Primarily, it is an M3 antagonist, inhibiting detrusor muscle contraction.
	Tolterodine	Phenylpropanolamine scaffold; hydroxylated aryl and diisopropylamine	Low	Overactive bladder; urinary urgency	M2/M3 antagonist; reduces bladder contractions
	Trihexyphenidyl (Benzhexol)	Cyclohexyl phenyl tertiary alcohol and piperidine	High	Parkinsonism; antipsychotic-induced EPS	Central M1 antagonist; modulates basal ganglia cholinergic tone

**Table 4 toxins-17-00469-t004:** Summary of pharmacological profiles and clinical relevance of key compounds.

Compound	Structural Class	Key Activities	Target Pathways/Mechanisms	Therapeutic Applications
Harmane	β-Carboline alkaloid	Antioxidant, neuroprotective, MAO inhibitor, tremor-inducing	MAO-A/B inhibition, serotonin modulation	Parkinson’s, Alzheimer’s, depression, HIV2
Norharmane	β-Carboline alkaloid	Antidepressant, anticancer, MAO inhibitor, photosensitizer	MAO-A/B inhibition, PIN transport inhibition	Depression, cancer, neurodegeneration4
Daturafolisides A–I	Steroidal glycosides (withanolides)	Anti-inflammatory, immunosuppressive, cytotoxic	NF-κB, MAPK, COX-2, and iNOS inhibition	Psoriasis, cancer, and autoimmune disorders
Daturataturin A and B	Withanolide glycosides	Anti-inflammatory, autophagy-inducing, immunosuppressive, cytotoxic	PI3K-Akt-mTOR, ERK/JNK/p38, NF-κB	Psoriasis, cancer, chronic inflammation5
Withametelins I–P	Withanolide-type lactones	Cytotoxic, anti-inflammatory, antioxidant, and kinase inhibition	NF-κB, MAPK, CDKs, Nrf2/Keap1/HO-1	Cancer, neuroinflammation, and autoimmune diseases
Daturametelin J	Withanolide lactone glycoside	Anti-inflammatory, cytotoxic, antioxidant, neuroprotective	NF-κB, COX-2, iNOS, Nrf2/HO-1	Psoriasis, cancer, Alzheimer’s, Parkinson’s

## Data Availability

No new data were created or analyzed in this study.

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
