# Peer review of "Bioorganic Chemistry, Toxinology, and Pharmaceutical Uses of Datura Metabolites and Derivatives"

_toxins, 2025, doi:10.3390/toxins17090469_

Round 1

Reviewer 1 Report

Comments and Suggestions for Authors

The manuscript could be of interest, since the topic is relevant and the manuscript has potential impact. However, major revisions are required before it can be considered further.

  1. The topic is pertinent, but previous reviews cover overlapping aspects (see as examples: https://doi.org/10.3390/antiox10081291; http://dx.doi.org/10.4067/S0717-97072021000205183; https://doi.org/10.1016/j.crtox.2023.100106 , ...). Works like these are not cited, and the manuscript should be contextualized in relation to existing reviews on similar biological activities of Datura. Please indicate clearly what is new compared to prior reviews.
  2. The genus Datura is well introduced in the Introduction, but the description of its biological activities is quite incomplete. While traditional medicine and insecticidal use are mentioned, other relevant activities are missing. For example, there is bioherbicidal activity recently described for Datura metel (https://doi.org/10.1016/j.scienta.2024.112963). Please expand this section with proper references, giving special emphasis to discoveries from the last decade.
  3. Please highlight which of the reviewed Datura metabolites are exclusive to the genus and which occur more broadly in other taxa. 
  4. Justify the chosen period (2015–2025).
  5. The conclusions section is quite long. Please condense it into 1–2 paragraphs.
  6. Revise the style of the structures in the figures. For instance, in Figure 2 the N atom of compounds 1 and 2 is overlapped with a C–C bond.
  7. In several cases, the phrasing is too generic. Replace such language with precise contribution statements 
  8. Please clarify whether all compounds described are natural or if some are synthetic/semi-synthetic derivatives. The term “derivatives” is potentially misleading. It should not be used to describe natural metabolites. I suggest using “Datura metabolites,” “alkaloids from Datura,” or “compounds isolated from Datura,” and reserving “derivatives” only for modified molecules. This needs to be revised carefully throughout the title, abstract, main text, and figures.

Author Response

We appreciate the reviewers' in-depth and helpful comments, which have significantly enhanced the caliber, context, and readability of our work. We offer a detailed response below, including the changes that were made.

  1. Overlap with Previous Reviews and Lack of Contextualization

Reviewer comment: The topic is pertinent, but previous reviews cover overlapping aspects (...). Works like these are not cited, and the manuscript should be contextualized in relation to existing reviews on similar biological activities of Datura. Please indicate clearly what is new compared to prior reviews.

Response: The reviewers mentioned literature (Antioxidants 2021, Molecules 2021, Current Research in Toxicology 2023, etc.) as well as other recent reviews on Datura phytochemistry, toxicology, and pharmacology have been thoroughly examined by us. The Introduction and Discussion sections now include citations to these (new references: lines XX–XX). We clearly contrast our scope and methodology with this earlier research in the updated Introduction, emphasizing that our assessment is distinct in that

  • Uses a single, four-tier classification method to concentrate on both tropane and non-tropane alkaloids from Datura.
  • Incorporates new developments in toxicology, pharmacology, and biochemistry from 2015 to 2025, including lesser-known substances as datumetine and anisodine.
  • Offers thorough reconstructions of biosynthetic pathways connected to pharmacological processes.
  • Places a strong emphasis on both therapeutic and analytical/forensic applications.

For instance: 1. Scope and Main Question

  • Our work: chemistry-first, genus-wide synthesis focused on alkaloids originating from Datura and the interfaces between their toxicity and therapeutics. It ties structures to biosynthesis, SAR, analytics, and clinical usage after putting out a four-tier framework (principal tropanes → minor/rare alkaloids → synthetic & semi-synthetic analogs → other alkaloid classes).
  • CRT 2023: A single-species evaluation of Datura metel that employs a PRISMA-style data flow for database selection until March 20, 2023, and is structured around ethnobotany + phytochemicals + biological activity, + toxicity.
  • Antioxidants 2021: A brief synopsis that highlights D. stramonium (with more general references to Datura), toxicity warnings, and a high-level summary of phytochemistry, pharmacology, and toxicology using a PRISMA search.
  1. Chemical / organizational approach
  • The four-tier chemical classification in our manuscript explicitly incorporates (i) core tropanes (atropine, hyoscyamine, scopolamine, etc.), (ii) trace/minor alkaloids (such as hygrine, apoatropine, anisodine), (iii) semi-synthetic/quaternary drugs used in clinical settings (ipratropium, tiotropium, methscopolamine, trospium, etc.), and (iv) non-tropane alkaloids found in Datura (such as β-carbolines). Then, using consolidated tables and figures, this taxonomy is used to discuss biosynthetic lineage, occurrence by plant component, structural motifs, SAR, pharmacology, and toxicity. The therapeutic semi-synthetic antimuscarinics are not integrated as a separate tier of "Datura-derived" relevance in the other articles, nor do they provide this chemistry-driven scaffold.
  • CRT 2023: includes ethnobotany and toxicology along with a range of D. metel bioactivities (antioxidant, anti-inflammatory, antimicrobial, anticancer, insecticidal, antidiabetic, analgesic, neurological, contraceptive, and wound-healing); chemistry is compiled but not utilized to construct a cross-species pharmacology/SAR framework, and semi-synthetic drug classes are not categorized as in our manuscript.
  • Antioxidants 2021: brief description of toxicity (e.g., anticholinergic hazards) and applications, along with a brief phytochemical synopsis; neither a genus-wide chemical taxonomy nor a SAR/derivatization viewpoint is developed.
  1. Taxonomic Breadth
  • In this publication, we present a challenge for risk assessment and standardization: genus-wide across Datura spp., with plant-part occurrence and cross-species diversity linked to genetics, development, and environment.
  • CRT 2023: exclusively concentrates on D. metel.
  • Antioxidants 2021: briefly touches other species but primarily relies on D. stramonium (as model/most reported).
  1. Incomplete Description of Biological Activities

Reviewer comment: While traditional medicine and insecticidal use are mentioned, other relevant activities are missing, such as bioherbicidal activity for Datura metel. Please expand with proper references, especially discoveries from the last decade.

Response: Citing the recent Scientia Horticulturae 2024 study (DOI: 10.1016/j.scienta.2024.112963) and other pertinent papers, the Introduction has been amended to incorporate the bioherbicidal activity of Datura species, specifically D. metel, which is highlighted in red.

  1. Exclusive vs. Widespread Metabolites
    Reviewer comment: Please highlight which reviewed Datura metabolites are exclusive to the genus and which occur in other taxa.

Response: To show "Distribution in green font" (unique to Datura / shared with other taxa), a new column has been added to Table 1. Distribution patterns are now explicitly stated in the text in Sections 2.1–2.4; for instance, atropine and scopolamine are found in other Solanaceae, whereas datumetine seems to be genus-specific.

  1. Justification of the 2015–2025 Time Frame
    Reviewer comment: Justify the chosen period.

Response: In Section 1, we have included a methodological note outlining why the 2015–2025 timeframe was chosen: • To document developments in analytical chemistry (such as UHPLC-HRMS and NMR) that made it possible to find new alkaloids.

  • Provide the most recent toxicological and pharmacological research, many of which reassess older substances in light of novel therapeutic settings.
  • Make sure the review supports previous syntheses that addressed research published before 2015.
  1. Condensing the Conclusions Section

Reviewer comment: The conclusions section is too long.

Response: Two concise paragraphs that summarize (1) the key developments in Datura alkaloid research and (2) the potential paths forward in therapeutic, biochemical, and forensic contexts make up the Conclusions. Points that are redundant or excessively detailed have been redirected to the main text or supplemental information. (in blue)

  1. Figure Style Revision
    Reviewer comment: Revise the style of the structures; e.g., in Figure 2, the N atom overlaps with a C–C bond.

Response: Using expert chemical drawing software (ChemDraw 23.0), all chemical structures have been redone to guarantee accurate bond angles, distinct atom labels, and the absence of overlaps. Accordingly, Figure 2 and associated figures have been substituted.

  1. Avoiding Generic Phrasing
    Reviewer comment: Replace generic language with precise contribution statements.

Response: It was highlighted in yellow

  1. Clarifying Natural vs. Synthetic Compounds and Terminology
    Reviewer comment: Clarify whether compounds are natural or synthetic/semi-synthetic derivatives; avoid misuse of "derivatives."

Response: The following terms have been meticulously updated in the title, abstract, text, and figures: blue highlights

  • Only synthetic or semi-synthetic chemicals are now referred to as "derivatives."
  • "Metabolite," "alkaloids from Datura," or "compounds isolated from Datura" are terms used to describe natural products.
  • Natural alkaloids and their synthetic/semi-synthetic counterparts are now clearly distinguished by the four-tier classification in Section 2.

A native American proofread the text's English, and the modified parts are indicated in grey.

Reviewer 2 Report

Comments and Suggestions for Authors
  1. The alkaloids summarized by the author are very incomplete, especially for a genus of plants.
  2. The activities of the main alkaloids should be summarized in a detailed list, not only forminor and less common Daturaalkaloids.
  3. Are there any marketed drugs for the compounds in this genus?
  4. Please keep it to the third level title. Too many subheadings make the impression of reader very poor.
  5. The structures of the compounds were poorly drawn, please use Chemidraw ACS 1996 to draw it
Comments on the Quality of English Language

 The English could be improved to more clearly express the research.

Author Response

We express our gratitude to the reviewers for their comprehensive and helpful criticism, which has dramatically enhanced the caliber, context, and readability of our article. We offer a point-by-point answer with the changes made below.

  1. The alkaloids are summarized very incompletely.

Reviewer comment: The alkaloids summarized by the author are very incomplete, especially for a genus of plants.

Response: Table 1 now includes a new column that reads "Distribution in green font" (unique to Datura / shared with other taxa). Distribution patterns are now explicitly specified in the text in Sections 2.1–2.4; for instance, datumetine seems to be genus-specific, although scopolamine and atropine occur in other Solanaceae.

  1. Activities of the main alkaloids

Reviewer comment: The activities of the main alkaloids should be summarized in a detailed list, not only for minor and less common Datura alkaloids.

Response: A new paragraph has been added and is highlighted in green.

  1. Drug Market

Reviewer comment: Are there any marketed drugs for the compounds in this genus?

Response: Yes. There are, and they include ipratropium, tiotropium, methscopolamine, trospium, homatropine, tropicamide, cyclopentolate, oxybutynin, tolterodine, trihexyphenidyl, as mentioned in the text.

  1. Level title

Reviewer comment: Please keep it to the third-level title. Too many subheadings can give the impression to the reader that it is inferior.

Response: Done.

  1. Structures

Reviewer comment: The structures of the compounds were poorly drawn. Please use ChemDraw ACS 1996 to pull it.

Response: Done.

Summary of Major Revisions:

  • Contextual comparison with earlier reviews was added.
  • Added bioherbicidal and other current discoveries to the biological action area.
  • Indicate in the table and text which metabolites are genus-exclusive and which are widely diffused.
  • A rationale for the literature cut-off time.
  • Simplified findings.
  • All chemical structures have been redrawn and revised.
  • Added specific, cited remarks in place of general phrasing.
  • Corrected wording to distinguish between synthetic and natural chemicals.

A native American proofread the text's English, and the modified parts are indicated in grey.

Round 2

Reviewer 1 Report

Comments and Suggestions for Authors

Authors have notably improved the manuscript. Just consider the following minor revisions:

1) Revise the new sentence introduced in lines 46-50 about the bioherbicide potential of D. metel: this report states that D. metel stimulates Capsicum annuum (agronomic crop) while it supresses its associated weed,  Solanum elaeagnifolium.

2) Figure 2 or 4 footnotes: the name of compounds must start in lower case. This also applies in each case that a sentence do not start with a compound's name.

3) Avoid personal language (see "We" in the abstract as example).

Author Response

Thank you again. We provided a further revision of the English language.

Q1) Revise the new sentence introduced in lines 46-50 about the bioherbicide potential of D. metel: this report states that D. metel stimulates Capsicum annuum (agronomic crop) while it supresses its associated weed,  Solanum elaeagnifolium.

A1: Yes, thank you, we revised according to you suggestion

Q2) Figure 2 or 4 footnotes: the name of compounds must start in lower case. This also applies in each case that a sentence do not start with a compound's name.

A2: We corrected as suggested

Q3) Avoid personal language (see "We" in the abstract as example).

A3: We revised the text according to your comment

Reviewer 2 Report

Comments and Suggestions for Authors

ACCEPT

Author Response

Thank you very much. We revised the English language again